# On the Integration of Spatial-Temporal Knowledge: A Lightweight Approach to Atmospheric Time Series Forecasting

**Yisong Fu**[1,2], **Fei Wang**[1,2*], **Zezhi Shao**[1], **Boyu Diao**[1,2], **Lin Wu**[1,2],
**Zhulin An**[1,2], **Chengqing Yu**[1,2], **Yujie Li**[1,2], **Yongjun Xu**[1,2*]

[1]State Key Laboratory of AI Safety, Institute of Computing Technology,
Chinese Academy of Sciences    [2]University of Chinese Academy of Sciences
{fuyisong24s, wangfei, shaozezhi, diaoboyu, wulin}@ict.ac.cn,
{anzhulin, yuchengqing22b, liyujie23s, xyj}@ict.ac.cn

## Abstract

Transformers have gained attention in atmospheric time series forecasting (ATSF) for their ability to capture global spatial-temporal correlations. However, their complex architectures lead to excessive parameter counts and extended training times, limiting their scalability to large-scale forecasting. In this paper, we revisit ATSF from a theoretical perspective of atmospheric dynamics and uncover a key insight: spatial-temporal position embedding (STPE) can inherently model spatial-temporal correlations even without attention mechanisms. Its effectiveness arises from the integration of geographical coordinates and temporal features, which are intrinsically linked to atmospheric dynamics. Based on this, we propose **STELLA**, a **S**patial-**T**emporal knowledge **E**mbedded **L**ightweight mode**L** for **A**STF, utilizing only STPE and an MLP architecture in place of Transformer layers. With *10k* parameters and one hour of training, STELLA achieves superior performance on five datasets compared to other advanced methods. The paper emphasizes the effectiveness of spatial-temporal knowledge integration over complex architectures, providing novel insights for ATSF. The code is available at https://github.com/GestaltCogTeam/STELLA.

## 1 Introduction

Atmospheric time series forecasting (ATSF), such as weather forecasting and air quality prediction, is of great significance in a wide variety of domains such as agriculture, energy, and economics. In recent decades, automatic weather stations have grown exponentially, becoming a cornerstone of modern meteorology [34]. These stations are cost-effective for applications [3, 36] and are ideally positioned to provide a large volume of data to advance deep learning (DL) approaches in ATSF [27].

However, the application of DL in ATSF faces two main challenges: (1) the observations of worldwide stations exhibit intricate spatial-temporal correlations [40], necessitating models with advanced mining capabilities to ensure accurate forecasting [52]. (2) The requirement for fine-grained and large-scale forecasting [25, 39] calls for highly efficient and scalable models.

These two challenges are often *trade-offs* in prior studies, as shown in Figure 1. Transformer and its variants, which have gained significant popularity in ATSF, utilize sophisticated architectures to capture global spatial-temporal correlations. For example, AirFormer [16] introduces dartboard attention for air quality prediction, and MRIformer [47] utilizes multi-resolution attention to predict wind

---

*Corresponding authors.

39th Conference on Neural Information Processing Systems (NeurIPS 2025).

speed. However, these complex designs come with significant costs, including hundreds of millions of parameters and extended training times, which limit their scalability for large-scale forecasting and hinder their applicability, especially with limited computational resources [6]. Moreover, despite the increased complexity, the performance gains are quite limited and do not justify the trade-off for practical utility. This motivates us to rethink the bottleneck of ATSF.

To this end, we delve deeper into the physical principles of atmospheric dynamics. In the atmospheric system, the evolution of atmospheric variable $\nu$ can be described by a partial differential equation (PDE):

$$\frac{\partial \nu}{\partial t} = f\left(\nu, t, \lambda, \phi, z\right). \qquad (1)$$

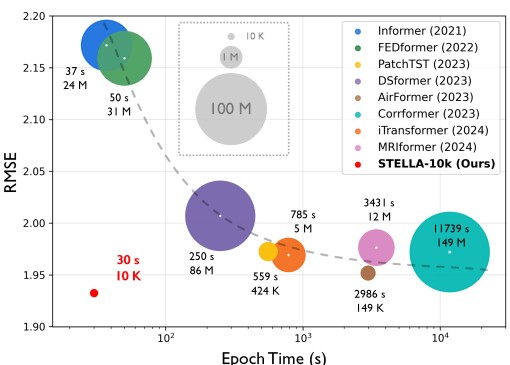

We demonstrate it theoretically in §3.5. The equation indicates that $\nu$ is directly influenced by time $t$ and geographical coordinates $\lambda, \phi, z$. However, most previous work has overlooked this spatial-temporal knowledge, simplistically treating $f$ as a function of historical values while focusing on fitting it in more complex forms. This misguided direction has led to the key bottleneck of ATSF.

In this paper, we highlight the significance of spatial-temporal knowledge and introduce a spatial-temporal position embedding (STPE) to integrate geographical coordinates and tempo-

Figure 1: Performance-efficiency comparison on the GlobalWind dataset. A performance-efficiency trade-off can be observed in Transformers, while STELLA leads in both aspects. The area of the plot represents the parameter count of the model.

ral features from Eq.(1). Although position embeddings are widely considered as an adjunct to permutation-invariant attention mechanisms, we demonstrate that **STPE can inherently model spatial-temporal correlations even in the absence of attention mechanisms, offering a '*free lunch*' in balancing the performance-efficiency trade-off.**

Consequently, we propose **STELLA**, a **S**patial-**T**emporal knowledge **E**mbedded **L**ightweight mode**L** for **A**STF. STELLA utilizes STPE and replaces the Transformer layers with a simple MLP. Figure 1 shows STELLA's lead in both performance and efficiency. With only *10k* parameters and one hour of training, STELLA achieves competitive performance against 17 baselines. Furthermore, it is noteworthy that the computational complexity of STELLA grows linearly with the increase of the number of stations $N$ and the parameter count is independent of $N$. Therefore, STELLA can efficiently scale to the data with a larger $N$, facilitating large-scale forecasting. STELLA's leading-edge performance and efficiency challenge the prevailing assumptions that ATSF necessitates complex architecture (e.g., Transformers, STGNNs), offering a balanced solution for ATSF. Our contributions can be summarized as follows.

- We innovatively highlight the significance of STPE in Transformer-based ATSF models. Even without attention mechanisms, it can explicitly capture spatial-temporal correlations by integrating spatial-temporal knowledge into the model. We theoretically prove its effectiveness from the perspective of atmospheric dynamics. Furthermore, STPE can also be applied to other models to improve performance (§4.5).
- We propose STELLA, utilizing the STPE and replacing the Transformer layers with a simple MLP. To the best of our knowledge, it is the first lightweight model designed for ATSF, challenging the prevailing assumption that ATSF necessitates complex architecture.
- STELLA offers a performance-efficiency balanced solution for ATSF. Extensive experiments across five datasets demonstrate the superior performance of STELLA over 17 baselines.

## 2 Related Work

### 2.1 DL in Atmospheric Time Series Forecasting

Atmospheric time series forecasting (ATSF), including applications such as weather forecasting and air quality prediction, involves predicting key atmospheric variables (e.g., temperature) collected from

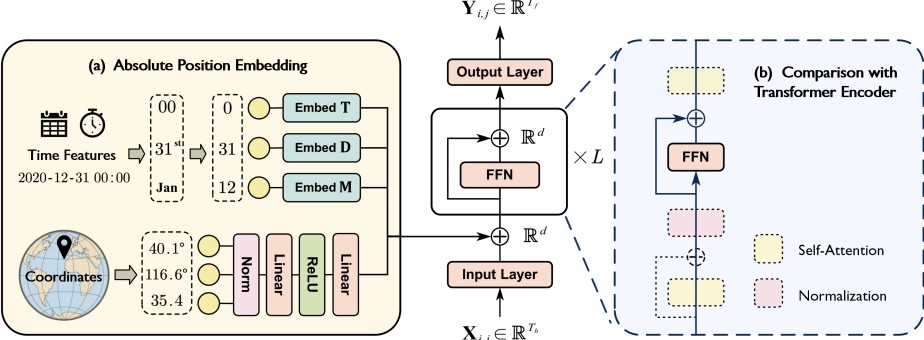

Figure 2: Architecture of STELLA.

weather stations over time. Spatial-temporal graph neural networks (STGNNs) prove effective in modeling spatial-temporal patterns across global stations [17, 22, 32, 1, 41, 28, 44], but most models are limited to short-term forecasting due to their high computational complexity, which restricts their scalability. Recently, Transformer-based models have gained popularity for their ability to capture global spatial-temporal correlations. For example, AirFormer [16] introduces dartboard attention to model spatial correlations in air quality prediction, while Corrformer [40] uses a multi-correlation mechanism as a substitute for attention in weather forecasting. MGSFformer [46] and MRIformer [47] employ attention to capture multi-resolution correlations through downsampling, aiming to forecast wind speed and air quality. However, the complexity of these models introduces significant computational costs, limiting their practical applicability and scalability for large-scale forecasting.

## 2.2 Studies of the effectiveness of Transformers

The effectiveness of Transformers has been thoroughly discussed in the fields of computer vision (CV) [48, 19] and natural language processing (NLP) [4]. In time series forecasting (TSF), LSTF-Linear [49] pioneered the exploration and outperformed a variety of Transformer-based methods with a linear model. Shao et al. [30] posited that Transformer-based models face an over-fitting problem on specific datasets. Some recent works further questioned the necessity of attention in Transformers for TSF and replaced the attention with other modules. For example, MTS-Mixers [14] attempt to use random matrices and factorized MLPs instead of attention for information aggregation. MEAformer [10] replaces conventional attention with a linear-complexity mixing module. Additionally, SOFTS [7] employs STAR module as a substitute for attention, which aggregates all series into a global core representation. These studies consider PE as supplementary to attention mechanisms and consequently remove it along with attention, yet none have recognized the importance of PE.

## 3 Methodology

### 3.1 Problem Definition

**Atmospheric Time Series Forecasting.** We consider $N$ stations and each station collects $C$ time series of atmospheric variables (e.g., temperature). Then the observed data at time $t$ can be denoted as $\mathbf{X}_t \in \mathbb{R}^{N \times C}$. The 3D geographical coordinates of stations are organized as a matrix $\mathbf{\Sigma} \in \mathbb{R}^{3 \times N}$, which is naturally accessible in station-based forecasting. Given the historical time series of all stations from the past $T_h$ time steps and optional spatial and temporal information, we aim to learn a function $\mathcal{F}_\theta(\cdot)$ to forecast the values of future $T_f$ time steps :

$$\mathbf{Y}_{t:t+T_f} = \mathcal{F}_\theta(\mathbf{X}_{t-T_h:t}; \mathbf{\Sigma}, t), \tag{2}$$

where $\mathbf{X}_{t-T_h:t} \in \mathbb{R}^{T_h \times N \times C}$ is the historical data, and $\mathbf{Y}_{t:t+T_f} \in \mathbb{R}^{T_f \times N \times C}$ is the future data.

### 3.2 Overview of STELLA

As shown in Figure 2, STELLA consists of an STPE module (Figure 2a) and an encoder that retains only the Feed-Forward Network (FFN) and discards all other components (Figure 2b). This

design significantly enhances computational and memory efficiency while maintaining competitive performance relative to more complex architectures. Our choice to pair STPE with a simple MLP architecture was driven by a key motivation: to isolate and demonstrate the standalone power of STPE in spatial-temporal modeling. This minimalistic yet effective design not only reduces computational overhead but also offers clear interpretability regarding the contribution of spatial-temporal knowledge in ATSF.

### 3.3 Spatial-Temporal Position Embedding

Position embedding encodes the positional information of tokens in a sequence [38] and is widely regarded as an auxiliary component for permutation-invariant attention mechanisms. However, we introduce spatial-temporal position embedding (STPE) to integrate geographical coordinates and temporal features into the model, demonstrating that it can inherently capture spatial-temporal correlations by embedding additional spatial-temporal knowledge. Specifically, STPE consists of two components: spatial embedding and temporal embedding.

**Spatial Embedding.**  The spatial embedding provides the geographical coordinates of stations to the model, which can explicitly model spatial correlations among worldwide stations. Specifically, we encode the geographical coordinates of the station into latent space. First, to account for the differing ranges of coordinate values, we perform normalization on each coordinate independently. Then, we utilize a feed-forward network (FFN). Therefore, the spatial embedding $\mathbf{SE}^i \in \mathbb{R}^d$ can be denoted as:

$$\mathbf{SE}^i = \text{FFN}(\mathbf{\Sigma}^i) = \mathbf{W}_2 \text{ReLU}\left(\mathbf{W}_1 \mathbf{\Sigma}^i + b_1\right) + b_2, \tag{3}$$

where $\mathbf{\Sigma}^i \in \mathbb{R}^3$ represents the normalized coordinates of station $i$.

**Temporal Embedding.**  Temporal embedding provides real-world temporal knowledge to the model. We utilize three learnable embedding matrices $\mathbf{T} \in \mathbb{R}^{24 \times d}$, $\mathbf{D} \in \mathbb{R}^{31 \times d}$ and $\mathbf{M} \in \mathbb{R}^{12 \times d}$ to save the temporal embeddings of all time steps [31]. They represent the patterns of weather in three scales ( $\mathbf{T}$ denotes hours in a day, $\mathbf{D}$ denotes days in a month and $\mathbf{M}$ denotes the months in a year), contributing to modeling the multi-scale temporal correlations of atmospheric states. We add them together to obtain temporal embedding $\mathbf{TE}_t$:

$$\mathbf{TE}_t = \mathbf{T}_t + \mathbf{D}_t + \mathbf{M}_t. \tag{4}$$

### 3.4 MLP Backbone

**Input Layer.**  Let $\mathbf{X}^{i,j} \in \mathbb{R}^{T_h}$ be the historical time series of station $i$ and variable $j$. $\mathbf{X}^{i,j}$ is mapped by the input embedding layer to $\mathbf{H}^{i,j} \in \mathbb{R}^d$ in latent space, then added to the spatial and temporal embedding to obtain $\mathbf{E}_t^{i,j}$:

$$\begin{aligned} \mathbf{H}^{i,j} &= \text{Linear}(\mathbf{X}^{i,j}), \\ \mathbf{E}_t^{i,j} &= \mathbf{H}^{i,j} + \mathbf{SE}^i + \mathbf{TE}_t. \end{aligned} \tag{5}$$

**Encoder.**  We utilize an $L$-layer MLP as encoder to learn the representation $\mathbf{Z}^{i,j}$ from the embedded data $\mathbf{E}_t^{i,j}$. Let $(\mathbf{Z}^{i,j})^0 = \mathbf{E}_t^{i,j}$, and the $l$-th MLP layer with residual connection can be denoted as:

$$(\mathbf{Z}^{i,j})^{l+1} = \text{FFN}^l\left((\mathbf{Z}^{i,j})^l\right) + (\mathbf{Z}^{i,j})^l. \tag{6}$$

**Output Layer.**  We employ a linear layer to map the representation $\mathbf{Z} \in \mathbb{R}^{d \times N \times C}$ to the specified dimension, generating the prediction $\hat{\mathbf{Y}} \in \mathbb{R}^{T_f \times N \times C}$.

### 3.5 Theoretical Analysis

In this part, we provide a theoretical analysis of STELLA, focusing on its effectiveness and efficiency.

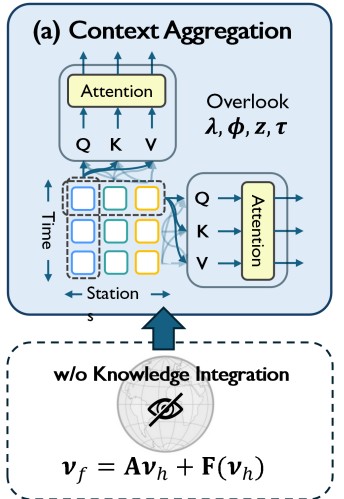 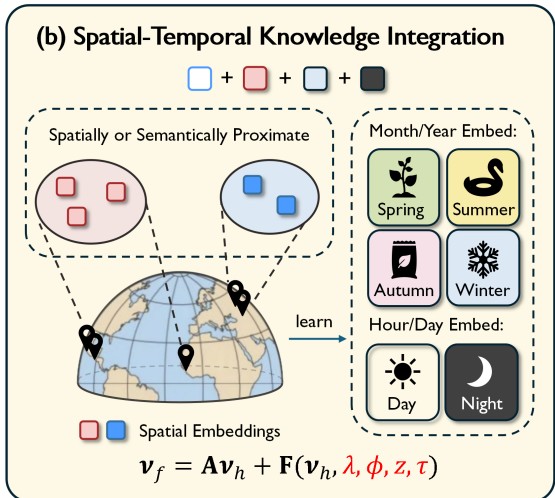

Figure 3: The key distinction that enhances the effectiveness of STELLA compared to prior approaches. (a) Most prior methods primarily rely on the context aggregation of historical observations (vanilla attention as an example). (b) In contrast, STELLA makes predictions guided by spatial and temporal knowledge. For clarity, history data is denoted as $\boldsymbol{\nu}_h$, future data as $\boldsymbol{\nu}_f$, and the day-in-month embedding is omitted.

**Effectiveness of STELLA.** The effectiveness of STELLA lies in the fact that STPE integrates geographical coordinates and real-world temporal knowledge into the model, which are intrinsically linked to atmospheric dynamics. In the following, we theoretically demonstrate this relationship.

**Theorem 1.** *Let $\{\lambda, \phi, z\}$ be the longitude, latitude, and altitude of a weather station and $\nu$ be a meteorological variable collected by the station, then the time evolution of $\nu$ is a function of $\nu$, time $t$ and coordinate $\lambda, \phi, z$:*

$$\frac{\partial \nu}{\partial t} = f(\nu, \lambda, \phi, z, t). \tag{7}$$

*Proof.* We provide proof with zonal wind speed as an example. See Appendix A.1 for the full proof. According to the fundamental equations of atmospheric dynamics, the wind speed $\mathbf{V}$ satisfies:

$$\frac{d\mathbf{V}}{dt} = -\frac{1}{\rho}\nabla p - 2\boldsymbol{\Omega} \times \mathbf{V} + \mathbf{g} + \mathbf{F}, \tag{8}$$

where $p$ is the pressure, $\rho$ is the air density, and other terms are constants. We can transform the equation into spherical coordinates and apply the thin-layer approximation. The zonal wind speed $u$ can be expressed as:

$$\frac{du}{dt} = -\frac{1}{\rho}\frac{\partial p}{a\cos\phi\partial\lambda} + fv + \frac{uv\tan\phi}{a} + F_\lambda, \tag{9}$$

where $\frac{d}{dt} = \frac{\partial}{\partial t} + u\frac{\partial}{a\cos\phi\partial\lambda} + v\frac{\partial}{a\partial\phi} + w\frac{\partial}{\partial z}$, $a$ is the Earth's radius. It is possible to render the left side of the equation spatially independent by rearranging terms:

$$\frac{\partial u}{\partial t} = -\left(u\frac{\partial u}{a\cos\phi\partial\lambda} + v\frac{\partial u}{a\partial\phi} + w\frac{\partial u}{\partial z}\right) - \frac{1}{\rho}\frac{\partial p}{a\cos\phi\partial\lambda} + fv + \frac{uv\tan\phi}{a} + F_\lambda. \tag{10}$$

Therefore, we have

$$\frac{\partial u}{\partial t} = f\left(u, \lambda, \phi, z, t\right). \tag{11}$$

$\square$

Considering using historical data spanning $T_h$ steps to predict future $T_f$ steps, we can derive the following corollary:

**Corollary 2.**
$$\boldsymbol{\nu}_{\tau+1:\tau+T_f} = \mathbf{A}\boldsymbol{\nu}_{\tau-T_h+1:\tau} + \mathbf{F}(\boldsymbol{\nu}_{\tau-T_h+1:\tau}, \lambda, \phi, z, \tau), \tag{12}$$

*where $\boldsymbol{\nu}_{\tau-T_h+1:\tau}$ is the historical data, $\boldsymbol{\nu}_{\tau+1:\tau+T_f}$ is the future data, and $||\mathbf{A}||_\infty = 1$.*

The detailed proof is provided in Appendix A.2. According to Eq.(12), we can employ a neural network $\mathcal{F}_\theta$ to approximate $\mathbf{F}(\boldsymbol{\nu}, \lambda, \phi, z, \tau)$. However, most previous models, including Transformers and STGNNs, have overlooked the critical spatial-temporal factors $\lambda, \phi, z, \tau$, instead treating $\mathbf{F}$ simplistically as a function of historical values, as shown in Figure 3 (a). These studies introduce increasingly complex context aggregation methods in an attempt to better fit historical data. However, blindly guessing spatial-temporal correlations by solely aggregating historical context may lead to overfitting, which becomes the key bottleneck for these models.

In comparison, STELLA can explicitly model $\mathbf{F}$ with $\lambda, \phi, z, \tau$ introduced by STPE, as shown in Figure 3 (b). STELLA aims to learn similar STPEs for spatially or semantically proximate stations and time steps, thereby capturing spatial-temporal correlations. Specifically, the spatial embedding learns the climatic features of different locations, while the temporal embedding captures the periodicity and seasonality of atmospheric states. We demonstrate this through visualization in §4.6. Therefore, STPE serves as an effective method for modeling spatial-temporal correlations.

**Efficiency of STELLA.** We theoretically analyze the efficiency of STELLA from the perspectives of parameter volume and computational complexity.

**Theorem 3.** *The total number of parameters required for STELLA is*
$$\underbrace{2(d+1)dL + (T_h + T_f + 2)d}_{MLP} + \underbrace{(d+72)d}_{STPE}. \tag{13}$$

The proof is provided in Appendix A.3. According to **Theorem 3**, the parameter count of STELLA is independent of the number of stations $N$. Therefore, a key advantage of STELLA is that its deployment overhead remains static regardless of how many stations it serves, making it particularly suitable for resource-constrained environments. In addition, the computational complexity of STELLA grows linearly with $N$. In contrast, context aggregation causes quadratic complexity and $\mathcal{O}(N)$ parameters, which are unaffordable with large-scale stations. Therefore, STELLA efficiently models spatial correlations with $N$-independent parameters and linear complexity, ensuring optimal scalability for large-scale data.

## 4 Experiments

### 4.1 Experimental Setup

**Datasets.** We conduct experiments on five real-world datasets: (1) **GlobalWind and GlobalTemp** [40] comprise the hourly averaged wind speed and temperature of 3,850 stations on a global scale, spanning two years. Following the prior work [40], we use the past 48 hours to predict the next 24 hours for short-term weather forecasting. (2) **ChinaWind and ChinaTemp** comprise the daily averaged wind speed and temperature of 396 stations in China, spanning 10 years. We use the past 60 days to predict the next 30 days, addressing long-term weather forecasting. (3) **China-PM2.5** comprises the hourly averaged wind speed of 1,316 stations in China, spanning five years. We use the past 72 hours to predict the next 72 hours for mid-term air quality prediction. See Appendix C.1 for more details of the datasets.

**Baselines.** We compare our STELLA with the following five categories of baselines. (1) *Classic methods*: HI [5], ARIMA [33]. (2) *STGNNs*: AGCRN [1], MTGNN [41], GTS [28]. (3) *Transformer-based TSF methods*: Informer [53], FEDformer [54], DSformer [45], PatchTST [23], iTransformer [20], DUET [26]. (3) lightweight TSF methods: N-BEATS [24], DLinear [49], FITS [43]. (4) *ATSF specialized Transformers*: AirFormer [16], Corrformer [40], MRIformer [47]. See Appendix C.2 for a detailed introduction to the baselines.

Table 1: Weather forecasting results on 5 datasets. The best results are in **bold** and the second best results are underlined. Dashes denote the out-of-memory (OOM) errors.

| Methods | GlobalWind | | GlobalTemp | | ChinaWind | | ChinaTemp | | ChinaPM2.5 | |
|---|---|---|---|---|---|---|---|---|---|---|
| | RMSE | MAE | RMSE | MAE | RMSE | MAE | RMSE | MAE | RMSE | MAE |
| Classic Methods | | | | | | | | | | |
| HI | 2.697 | 1.831 | 3.859 | 2.575 | 9.851 | 6.751 | 7.630 | 5.834 | 37.11 | 20.29 |
| ARIMA | 2.116 | 1.539 | 4.575 | 3.267 | 7.947 | 5.795 | 5.396 | 4.026 | 38.66 | 20.40 |
| STGNNs | | | | | | | | | | |
| AGCRN | - | - | - | - | 7.458 | 5.061 | 4.336 | 2.999 | - | - |
| MTGNN | - | - | - | - | 7.294 | 5.055 | 4.221 | 3.168 | 28.21 | 16.12 |
| GTS | - | - | - | - | 7.312 | 4.997 | 4.298 | 3.082 | - | - |
| Transformer-Based TSF Methods | | | | | | | | | | |
| Informer | 2.172 | 1.496 | 5.770 | 4.415 | 7.832 | 5.279 | 4.477 | 3.045 | 28.85 | 16.04 |
| FEDformer | 2.159 | 1.471 | 3.324 | 2.405 | 7.334 | 5.051 | 4.665 | 3.455 | 28.75 | 16.14 |
| DSformer | 2.007 | 1.347 | 3.089 | 2.057 | 7.311 | 5.000 | 4.919 | 3.605 | 27.68 | 14.94 |
| PatchTST | 1.973 | 1.332 | 3.130 | 2.062 | 7.295 | 5.033 | 4.909 | 3.600 | 26.87 | 14.37 |
| iTransformer | 1.969 | 1.314 | 2.950 | 1.883 | 7.248 | 4.995 | 4.292 | 3.193 | 26.82 | 14.40 |
| DUET | 1.946 | 1.318 | 3.072 | 2.042 | 7.278 | 5.029 | 4.480 | 3.048 | 26.61 | 14.41 |
| MLP-Based TSF Methods | | | | | | | | | | |
| N-BEATS | 2.031 | 1.390 | 3.034 | 2.117 | 7.297 | 4.998 | 4.791 | 3.486 | 26.48 | 14.44 |
| DLinear | 2.005 | 1.350 | 3.149 | 2.072 | 7.309 | 5.031 | 4.990 | 3.659 | 27.60 | 14.76 |
| FITS | 2.021 | 1.354 | 3.150 | 2.072 | 7.284 | 5.039 | 5.283 | 3.823 | 27.73 | 14.96 |
| ATSF Specialized Methods | | | | | | | | | | |
| AirFormer | 1.952 | 1.314 | 5.594 | 4.127 | 7.909 | 5.377 | 4.772 | 3.561 | 29.63 | 15.55 |
| Corrformer | 1.972 | 1.304 | 2.777 | 1.888 | 7.224 | 4.950 | 4.728 | 3.377 | 28.07 | 15.28 |
| MRIformer | 1.976 | 1.318 | 3.085 | 1.999 | 7.264 | 4.993 | 5.132 | 3.678 | 26.99 | 14.39 |
| STELLA-*10k* | 1.933 ± 0.001 | 1.294 ± 0.002 | 3.041 ±0.002 | 2.009 ±0.002 | 7.118 ±0.003 | 4.876 ± 0.001 | 4.167 ±0.002 | 3.003 ±0.003 | 26.32 ±0.003 | 14.08 ±0.002 |
| STELLA$_{opt}$ | **1.919** ± 0.002 | **1.284** ± 0.002 | **2.724** ±0.003 | **1.858** ±0.002 | **7.104** ±0.010 | **4.869** ± 0.001 | **4.112** ±0.003 | **2.975** ±0.005 | **26.15** ±0.004 | **13.85** ±0.001 |

**Implementation Details.** We develop STELLA-*10k* which has approximately 10k parameters. The number of MLP layers is 2, and the hidden dimension is 32. To explore the full performance of the model, we also conducted hyperparameter research on different datasets to find the optimal configuration, denoted as STELLA$_{opt}$. Detailed configurations are provided in Appendix C.4. We adopt the Adam optimizer [11] to train our model and the learning rate is set to 5e-4. We trained all baselines with MAE (Mean Absolute Error) loss [30, 15], and the results of all baselines are obtained using the best hyperparameters through hyperparameter search. We evaluate the performance of all baselines using two commonly used metrics: MAE and RMSE (Root Mean Square Error). All models are implemented with PyTorch 2.3.1 and trained on an NVIDIA GeForce RTX 4090 24GB GPU and an Intel Xeon Gold 6330 CPU.

## 4.2 Main Results

Table 1 presents the results of the performance comparison between STELLA and other baselines on all datasets. The results of STELLA are averaged over 5 runs, with the standard deviation included. It is evident that the performance of lightweight TSF methods, such as DLinear and N-BEATS, is not satisfactory. This indicates that lightweight TSF models, which fail to capture spatial-temporal correlations, are inadequate for large-scale prediction tasks. STGNNs suffer from OOM errors due to their high computational complexity, while the performance is also suboptimal. Similarly, despite the complex architectures of Transformer-based models, most of them exhibit limited performance.

In contrast, STELLA-*10k* achieves competitive performance with a simple MLP architecture and only 10k parameters, while STELLA$_{opt}$ consistently outperforms all other baselines on five datasets and three ATSF tasks. This suggests that integrating spatial-temporal information can significantly enhance model performance, proving to be more effective than the complex architectures of Transformer-based models. In addition, we provide a comparative analysis between STELLA and numerical weather prediction (NWP) methods in Appendix D.1.

## 4.3 Efficiency Analysis

Figure 1 illustrates the performance-efficiency comparison with Transformers. Here we further compare STELLA and other baselines, evaluating parameter counts, epoch time, and GPU memory usage [18]. Experiments are conducted on the most challenging GlobalWind dataset. As shown in Table 2, STELLA surpasses other DL methods in terms of all three efficiency metrics. When compared to the ATSF specialized methods, STELLA demonstrates an order-of-magnitude improvement across all three efficiency metrics, being about $10\times$ to $10,000\times$ smaller, $100\times$ to $300\times$ faster, and $50\times$ memory-efficient, respectively. Additionally, due to its compact parameter size and simple computa-

Table 2: Efficiency metrics of STELLA and other Transformer-based methods on GlobalWind.

| METHODS | PARAMS | EPOCH TIME | MAX MEM. |
|---|---|---|---|
| Informer | 23.94M | 37s | 1.39GB |
| FEDformer | 31.07M | 50s | 1.63GB |
| DSformer | 85.99M | 250s | 13.6GB |
| PatchTST | 424.1K | 559s | 19.22GB |
| iTransformer | 4.55M | 785s | 16.61GB |
| AirFormer | 148.7K | 2986s | 14.01GB |
| Corrformer | 148.7M | 11739s | 18.41GB |
| MRIformer | 11.66M | 3431s | 12.69GB |
| **STELLA-*10k*** | **9.98K** | **30s** (141s CPU) | **792MB** |

tions, STELLA can be efficiently trained in a CPU environment. It requires only 141 seconds to train STELLA-*10k* for an epoch, making it well-suited for environments with limited computational resources. See Appendix D.2 for detailed efficiency experiments under limited resources.

## 4.4 Ablation Study

**Effects of STPE.** STPE is the key component of STELLA. To study its effects, we first conduct experiments on models with the spatial embedding and temporal embedding removed separately. Table 3 reveals that removing either embedding component leads to a decrease in MSE. This indicates that both spatial and temporal embeddings contribute positively to model performance. Next, we compare relative position embedding (RPE) with STPE. Specifically, RPE embeds the indices of stations instead of the absolute geographical coordinates. Since we project the temporal dimension into the hidden space, temporal RPE is unnecessary. The results are presented in Table 3. Although RPE introduces $Nd$ parameters, significantly increasing the model size, its performance still falls short of that of STPE, further validating its effectiveness of STPE.

Table 3: Ablation results of STELLA on five datasets.

| Methods | GlobalWind | | GlobalTemp | | ChinaWind | | ChinaTemp | | ChinaPM2.5 | |
|---|---|---|---|---|---|---|---|---|---|---|
| | RMSE | MAE | RMSE | MAE | RMSE | MAE | RMSE | MAE | RMSE | MAE |
| w/o Spatial Embedding | 1.962 | 1.329 | 2.821 | 1.921 | 7.199 | 4.953 | 4.798 | 3.506 | 26.32 | 13.95 |
| w/o Temporal Embedding | 1.963 | 1.327 | 2.812 | 1.918 | 7.209 | 4.952 | 4.357 | 3.163 | 26.63 | 14.08 |
| RPE | 1.960 | 1.341 | 2.900 | 1.984 | 7.197 | 4.925 | 4.555 | 3.340 | 26.50 | 13.96 |
| STELLA | **1.919** | **1.284** | **2.724** | **1.858** | **7.104** | **4.869** | **4.112** | **2.975** | **26.15** | **13.85** |

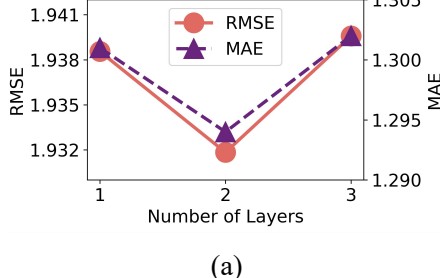

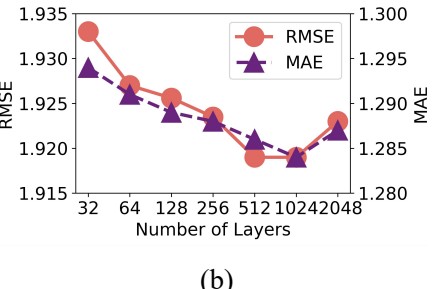

(a)                                        (b)

Figure 4: Results of hyperparameters analysis on GlobalWind dataset. (a) Effects of the number of layers ($d = 64$). (b) Effects of the hidden dimension ($L = 2$).

**Hyperparameter Study.** We investigate the effects of two important hyperparameters: the number of layers $L$ in the MLP and the hidden dimension $d$. As illustrated in Figure 4 (a), STELLA achieves

Table 4: Improvements obtained by the adoption of STPE.

| Datasets | | GlobalWind | | GlobalTemp | | ChinaWind | | ChinaTemp | |
| --- | --- | --- | --- | --- | --- | --- | --- | --- | --- |
| Metric | | RMSE | MAE | RMSE | MAE | RMSE | MAE | RMSE | MAE |
| PatchTST | Original | 1.973 | 1.332 | 3.130 | 2.062 | 7.295 | 5.033 | 4.909 | 3.600 |
| | **+STPE** | **1.947** | **1.307** | **3.077** | **1.994** | **7.196** | **4.978** | **4.490** | **3.303** |
| DSformer | Original | 2.007 | 1.347 | 3.089 | 2.062 | 7.311 | 5.000 | 4.919 | 3.605 |
| | **+ STPE** | **1.982** | **1.325** | **3.044** | **2.020** | **7.243** | **4.950** | **4.762** | **3.416** |
| iTransformer | Original | 1.969 | 1.314 | 2.950 | 1.883 | 7.248 | 4.995 | 4.292 | 3.193 |
| | **+STPE** | **1.917** | **1.279** | **2.855** | **1.804** | **7.104** | **4.914** | **4.188** | **3.600** |

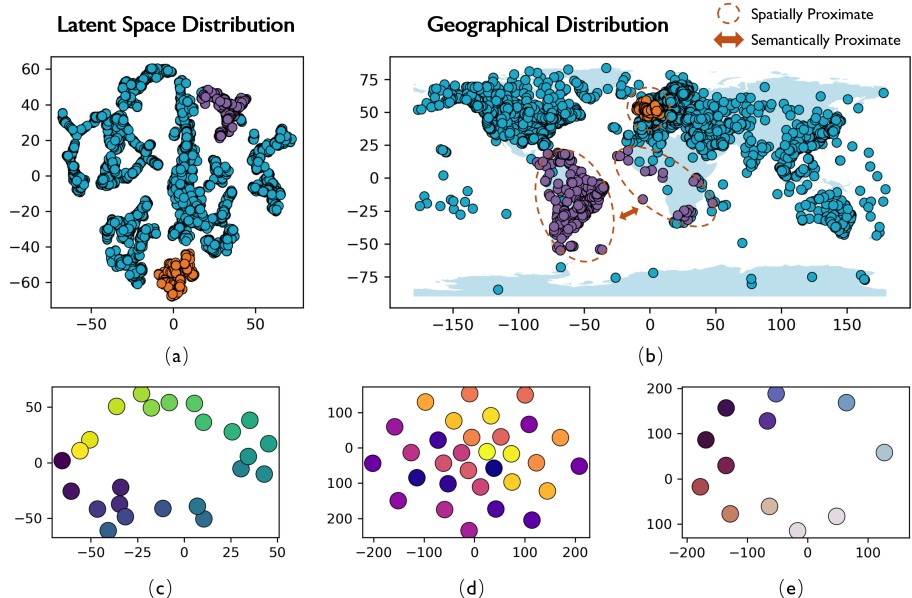

Figure 5: Visualization of STPE. (a) Spatial embedding in the 2D latent space. (b) Geographical distribution of stations. The model learns spatial and semantic similarity through spatial embedding. (c) Hour-in-day embedding $\mathbf{T}$. (d) Day-in-month embedding $\mathbf{D}$. (e) Month-in-year embedding $\mathbf{M}$.

the best performance when $L = 2$, whereas an increase in $L$ beyond 2 results in over-fitting and a consequent decline in model performance. Figure 4 (b) shows that the metrics decrease as the hidden dimension increases and begin to converge when $d$ exceeds 1024. However, STELLA-*10k* ($d = 32$) already outperforms other Transformer-based models. This further substantiates that STPE is more effective than the complex architectures of Transformer-based models.

## 4.5 Generalization of Spatial-Temporal Position Embedding

In this section, we further evaluate the effects of STPE by applying it to Transformer-based models, with the results reported in Table 4. Only Transformers that independently embed each channel are compatible with STPE (Informer, FEDformer, etc., are incompatible), as our approach generates spatial embeddings for each station individually. The result shows that STPE can significantly enhance the performance of Transformers, enabling them to achieve satisfactory results. In particular, iTransformer achieves state-of-the-art performance on GlobalWind after the application of STPE.

## 4.6 Visualization of PE

In this section, we visualize the STPE to further study its effectiveness. Due to the high dimensionality of the embeddings, we employ t-SNE [37] to visualize them on 2D planes.

**Visualization of Spatial Embedding.** Figure 5 (a) indicates that spatial embeddings tend to cluster. The model attempts to learn similar embeddings for spatially or semantically proximate stations, resulting in a clustered structure. To demonstrate this, we have marked two clusters with orange and purple and examined the geographical distribution of the stations within each cluster. As shown in Figure 5 (b), the orange cluster is densely distributed in Europe in the geographical space, indicating that the model has learned spatial similarity. Meanwhile, stations in the purple cluster are distributed across South America and Africa, with all distribution areas characterized by tropical or subtropical climates, suggesting that the model has captured semantic similarity.

**Visualization of Temporal Embedding.** Figure 5 (c-e) shows the temporal embeddings with colors representing the temporal order. The hour-in-day and month-in-year embeddings form ring-like structures in temporal order, revealing the distinct daily and annual periodicities of weather, which is consistent with humans' common understanding.

### 4.7 Case Study

To comprehensively illustrate STELLA's capability to capture the complex spatial-temporal correlations among large-scale stations, we present a visualization of the forecasting results of the Global-Wind dataset from a multi-station perspective. Additional illustrative showcases are available in Appendix F. Kriging [13] is employed to interpolate discrete points into a continuous surface, enhancing the visual clarity of spatial variations. As shown in Figure 6, the predicted results (right column) are closely aligned with the ground-truth values (left column) across all displayed time steps. This high level of consistency confirms that STELLA effectively captures spatial-temporal patterns in global weather data and delivers accurate predictions.

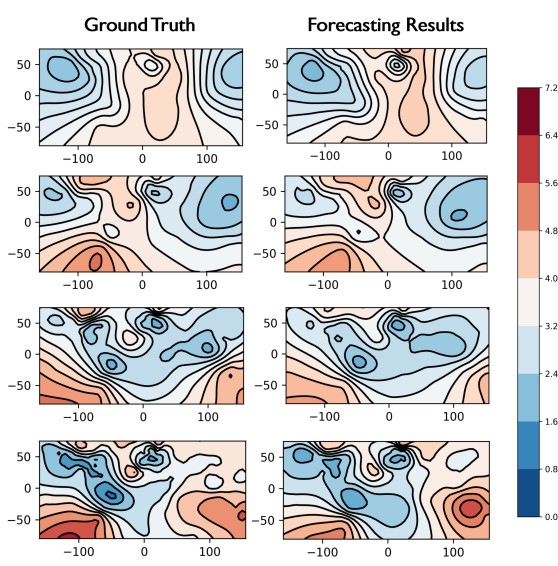

Figure 6: Forecasting results of averaged wind speed with a 6-hour interval and $5°$ (i.e., $64 \times 32$) resolution.

## 5 Conclusion

This work innovatively highlights the significance of STPE in Transformers for ATSF. Even without attention mechanisms, STPE explicitly captures spatial-temporal correlations by integrating geographical coordinates and temporal features, which are inherently linked to atmospheric dynamics. We then present STELLA, a lightweight and effective model for ATSF. We leverage STPE and replace Transformer layers with a simple MLP. STELLA can achieve satisfactory performance across five weather datasets. The paper posits that the incorporation of spatial-temporal knowledge is more effective than intricate model architectures, illuminating novel insights for ATSF.

**Limitations and Future Work.** STELLA is limited by its channel-independent modeling and the restricted expressive power of MLPs, and it may produce overly smoothed forecasts for extreme events, which is a common issue with MSE/MAE-trained models. Future work will address these challenges by incorporating covariates and physical constraints. See Appendix G for more details.

## Acknowledgement

This work is supported by the NSFC underGrant Nos.62372430 and 62502505, the Youth Innovation Promotion Association CAS No.2023112, the Postdoctoral Fellowship Program of CPSF under Grant Number GZC20251078, the China Postdoctoral Science Foundation No.2025M77154 and HUA Innovation fundings. We sincerely thank all the anonymous reviewers who gerenously contributed their time and efforts.

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

# A  Theoretical Proofs

## A.1  Full Proof of Theorem 1

*Proof.* The fundamental equations of atmospheric motion [21] are

$$
\begin{cases}
\dfrac{\mathrm{d}\mathbf{V}}{\mathrm{d}t} = -\dfrac{1}{\rho}\nabla p - 2\boldsymbol{\Omega} \times \mathbf{V} + \mathbf{g} + \mathbf{F}, \\[2mm]
\dfrac{\mathrm{d}\rho}{\mathrm{d}t} + \rho\nabla \cdot \mathbf{V} = 0, \\[2mm]
C_p\dfrac{\mathrm{d}T}{\mathrm{d}t} - \dfrac{1}{\rho}\dfrac{\mathrm{d}p}{\mathrm{d}t} = Q, \\[2mm]
\dfrac{\partial q}{\partial t} + \mathbf{V} \cdot \nabla q = S_q, \\[2mm]
p = \rho RT,
\end{cases}
\tag{14}
$$

where $\mathbf{V}$ is the velocity, $\rho$ is the air density, $p$ is the pressure, $T$ is the temperature, $q$ is the specific humidity, and others are physical constants.

Expand the equations into scalar form and transform them into spherical coordinates, yielding the following:

$$
\begin{cases}
\dfrac{\mathrm{d}u}{\mathrm{d}t} = -\dfrac{1}{\rho}\dfrac{\partial p}{r\cos\phi\,\partial\lambda} + fv + \dfrac{uv\tan\phi}{r} + F_\lambda, \\[2mm]
\dfrac{\mathrm{d}v}{\mathrm{d}t} = -\dfrac{1}{\rho}\dfrac{\partial p}{r\partial\phi} - fu - \dfrac{u^2\tan\phi}{r} + F_\phi, \\[2mm]
\dfrac{\mathrm{d}w}{\mathrm{d}t} = -\dfrac{1}{\rho}\dfrac{\partial p}{\partial r} - g + F_r, \\[2mm]
\dfrac{\mathrm{d}\rho}{\mathrm{d}t} = -\rho\left(\dfrac{1}{r\cos\phi}\dfrac{\partial u}{\partial\lambda} + \dfrac{\partial v}{r\partial\phi} + \dfrac{\partial w}{\partial r} + \dfrac{2w}{r} - \dfrac{v}{r}\tan\phi\right), \\[2mm]
\dfrac{\mathrm{d}T}{\mathrm{d}t} = \dfrac{Q}{C_p} + \dfrac{1}{\rho C_p}\dfrac{\mathrm{d}p}{\mathrm{d}t}, \\[2mm]
\dfrac{\mathrm{d}q}{\mathrm{d}t} = S_q, \\[2mm]
p = \rho RT,
\end{cases}
\tag{15}
$$

where $u$ is zonal velocity, $v$ is meridional velocity, and $w$ is vertical velocity. The expansion of $\frac{\mathrm{d}}{\mathrm{d}t}$ is

$$
\frac{\mathrm{d}}{\mathrm{d}t} = \frac{\partial}{\partial t} + u\frac{\partial}{r\cos\phi\,\partial\lambda} + v\frac{\partial}{r\partial\phi} + w\frac{\partial}{\partial r}.
\tag{16}
$$

Radial distance $r$ can be further denoted as $r = a + z$, where $a$ is Earth's radius and $z$ is altitude. Since $a$ is a constant and $a \gg z$, we have $\frac{\partial}{\partial r} = \frac{\partial}{\partial z}$ and we can approximate $r$ with $a$. Then we render the left side of the equation spatial independent by rearranging terms:

$$
\begin{cases}
\dfrac{\partial u}{\partial t} = -\dfrac{u}{a\cos\phi}\dfrac{\partial u}{\partial\lambda} - \dfrac{v}{a}\dfrac{\partial u}{\partial\phi} - w\dfrac{\partial u}{\partial z} - \dfrac{1}{\rho}\dfrac{\partial p}{a\cos\phi\partial\lambda} + fv + \dfrac{uv\tan\phi}{a} + F_\lambda, \\[2mm]
\dfrac{\partial v}{\partial t} = -\dfrac{u}{a\cos\phi}\dfrac{\partial v}{\partial\lambda} - \dfrac{v}{a}\dfrac{\partial v}{\partial\phi} - w\dfrac{\partial v}{\partial z} - \dfrac{1}{\rho}\dfrac{\partial p}{a\partial\phi} - fu - \dfrac{u^2\tan\phi}{a} + F_\phi, \\[2mm]
\dfrac{\partial w}{\partial t} = -\dfrac{u}{a\cos\phi}\dfrac{\partial w}{\partial\lambda} - \dfrac{v}{a}\dfrac{\partial w}{\partial\phi} - w\dfrac{\partial w}{\partial z} - \dfrac{1}{\rho}\dfrac{\partial p}{\partial z} - g + F_r, \\[2mm]
\dfrac{\partial\rho}{\partial t} = -\dfrac{u}{a\cos\phi}\dfrac{\partial\rho}{\partial\lambda} - \dfrac{v}{a}\dfrac{\partial\rho}{\partial\phi} - w\dfrac{\partial\rho}{\partial z} - \rho\left(\dfrac{1}{r\cos\phi}\dfrac{\partial u}{\partial\lambda} + \dfrac{\partial v}{r\partial\phi} + \dfrac{\partial w}{\partial r} + \dfrac{2w}{r} - \dfrac{v}{r}\tan\phi\right), \\[2mm]
\dfrac{\partial T}{\partial t} = -\dfrac{u}{a\cos\phi}\dfrac{\partial T}{\partial\lambda} - \dfrac{v}{a}\dfrac{\partial T}{\partial\phi} - w\dfrac{\partial T}{\partial z} + \dfrac{1}{\rho C_p}\left(\dfrac{\partial p}{\partial t} + \dfrac{u}{a\cos\phi}\dfrac{\partial p}{\partial\lambda} + \dfrac{v}{a}\dfrac{\partial p}{\partial\phi} + w\dfrac{\partial p}{\partial z}\right) + \dfrac{Q}{C_p}, \\[2mm]
\dfrac{\partial q}{\partial t} = -\dfrac{u}{a\cos\phi}\dfrac{\partial q}{\partial\lambda} - \dfrac{v}{a}\dfrac{\partial q}{\partial\phi} - w\dfrac{\partial q}{\partial z} + S_q, \\[2mm]
p = \rho R T.
\end{cases}
\tag{17}
$$

Therefore, for each atmospheric variable $\nu$, we have

$$
\frac{\partial\nu}{\partial t} = f(\nu, t, \lambda, \phi, z).
\tag{18}
$$

$\square$

## A.2 Proof of Corollary 2

*Proof.* According to **Theorem** 1 , we can integrate both sides of the equation with respect to $t$, from time step $\tau_i$ to step $\tau_j$:

$$
\nu_{\tau_j} = \nu_{\tau_i} + \int_{\tau_i\Delta t}^{\tau_j\Delta t} f\left(\nu(\lambda, \phi, z, t), \lambda, \phi, z, t\right)\mathrm{d}t,
\tag{19}
$$

where $\Delta t$ is the interval between time steps.

Before proceeding with the mathematical proof, we first illustrate with Figure 7. A directed edge from $\tau_i$ to $\tau_j$ in the figure denotes the evolution constrained by Eq.(19). Figure 7 (a) shows the mechanism that the state of the atmosphere evolves step by step, which we need to adopt an autoregressive neural network (e.g. RNN) to approximate it. Through a simple topological transformation, we can obtain the mechanism shown in (b), where the unobserved states are calculated by every historical observation. Therefore, we can adopt a neural network to predict all unobserved states in parallel and it constitutes a more robust approach as it fully leverages historical observations.

We provide a detailed mathematical proof in the following. For brevity, $\int_{\tau_i\Delta t}^{\tau_j\Delta t} F(\lambda, \phi, z, t)\mathrm{d}t$ is denoted as $I_i^j$. For the unobserved data at step $\tau + k$ ($k = 1, 2, \cdots, T_f$), it can be represented by:

$$
\begin{cases}
\nu_{\tau+k} = \nu_\tau + I_\tau^{\tau+k}, \\[1mm]
\nu_{\tau+k} = \nu_{\tau-1} + I_{\tau-1}^{\tau+k}, \\[1mm]
\quad\cdots \\[1mm]
\nu_{\tau+k} = \nu_{\tau-T_h+1} + I_{\tau-T_h+1}^{\tau+k}.
\end{cases}
\tag{20}
$$

We can take its linear combination as follows:

$$
\begin{aligned}
\nu_{\tau+k} &= \alpha_{k,1}\left(\nu_\tau + I_\tau^{\tau+k}\right) + \alpha_{k,2}\left(\nu_{\tau-1} + I_{\tau-1}^{\tau+k}\right) + \cdots + \alpha_{k,T_h}\left(\nu_{\tau-T_h+1} + I_{\tau-T_h+1}^{\tau+k}\right) \\
&= \boldsymbol{\alpha}_k\left(\boldsymbol{\nu}_{\tau-T_h+1:\tau} + \mathbf{I}_{\tau-T_h+1:\tau}^{\tau+k}\right),
\end{aligned}
\tag{21}
$$

where $\mathbf{I}_{\tau-T_h+1:\tau}^{\tau+k}$ is $(I_{\tau-T_h+1}^{\tau+k}, I_{\tau-T_h}^{\tau+k}, \cdots, I_\tau^{\tau+k})^\top$ and $\boldsymbol{\alpha}_k \in \mathbb{R}^{T_h}$ satisfies $||\boldsymbol{\alpha}_k|| = 1$.

By repeating this procedure from $\nu_{\tau+1}$ to $\nu_{\tau+T_f}$, we have

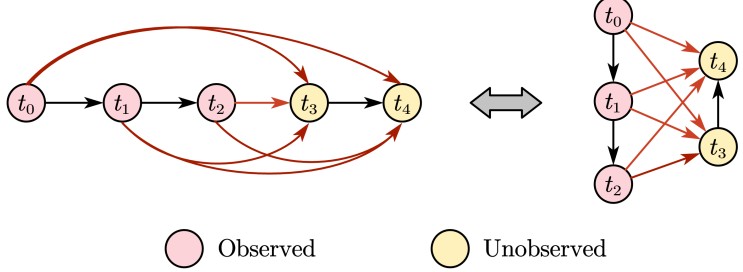

◯ Observed      ◯ Unobserved

Figure 7: Mechanism of the evolution of the atmosphere state. (a) The atmosphere state evolves step by step through the black edges. (b) Predict the unobserved states in parallel through the red edges. (a) and (b) are topologically equivalent.

$$\begin{cases} \boldsymbol{\nu}_{\tau+1} = \boldsymbol{\alpha}_1 \left( \boldsymbol{\nu}_{\tau-T_h+1:\tau} + \mathbf{I}^{\tau+1}_{\tau-T_h+1:\tau} \right), \\ \boldsymbol{\nu}_{\tau+2} = \boldsymbol{\alpha}_2 \left( \boldsymbol{\nu}_{\tau-T_h+1:\tau} + \mathbf{I}^{\tau+2}_{\tau-T_h+1:\tau} \right), \\ \qquad \cdots \\ \boldsymbol{\nu}_{\tau+T_f} = \boldsymbol{\alpha}_{T_f} \left( \boldsymbol{\nu}_{\tau-T_h+1:\tau} + \mathbf{I}^{\tau+T_f}_{\tau-T_h+1:\tau} \right). \end{cases} \tag{22}$$

Therefore, we have

$$\boldsymbol{\nu}_{\tau+1:\tau+T_f} = \mathbf{A}\boldsymbol{\nu}_{\tau-T_h+1:\tau} + \mathbf{F}(\boldsymbol{\nu}_{\tau-T_h+1:\tau}, \lambda, \phi, z, \tau), \tag{23}$$

where $\mathbf{A} = \left( \boldsymbol{\alpha}_1, \boldsymbol{\alpha}_2, \cdots, \boldsymbol{\alpha}_{T_f} \right)^\top$ satsifies $||\mathbf{A}||_\infty = 1$ and $\mathbf{F}$ is the combination of $\mathbf{I}$. $\qquad \square$

### A.3 Proof of Theorem 3

*Proof.* The data embedding layer maps the input data into the latent space with dimension $d$, thereby introducing $(T_h+1)d$ parameters. Analogously, the regression layer introduces $(T_f+1)d$ parameters. The parameter count of a $L$-layer MLP with residual connection is $2Ld(d+1)$.

For the STPE module, the spatial embedding costs $(3+1)d + d(d+1)$ parameters, and the temporal embedding costs $(24+31+12)d$ parameters.

Therefore, the total number of parameters required for STELLA is $2(d+1)dL + (T_h + T_f + 2)d + (d+72)d$. $\qquad \square$

## B  Overall Workflow of STELLA

The overall workflow of STELLA is provided in Algorithm 1.

## C  Experimental Details

### C.1  Dataset Description

To evaluate the comprehensive performance of the proposed model, we conduct experiments on five ATSF datasets with different temporal resolutions and spatial coverages including:

- **GlobalWind and GlobalTemp** [40] are collected from the National Centers for Environmental Information (NCEI)[2]. These datasets contain the hourly averaged wind speed and temperature of 3,850 stations around the world, spanning two years from 1 January 2019 to

---

[2]https://www.ncei.noaa.gov/data/global-hourly/access

**Algorithm 1** Overall workflow of STELLA.

---

1: **INPUT:** historical data $\mathbf{X} \in \mathbb{R}^{T_h \times N \times C}$, geographical coordinates $\boldsymbol{\Sigma} \in \mathbb{R}^{N \times 3}$, the first time step $t$
2: **OUTPUT:** forecasting result $\mathbf{Y} \in \mathbb{R}^{T_f \times N \times C}$
3: $\mathbf{X} = \mathbf{X}.\text{transpose}\,(1, -1)$     /* $\mathbf{X} \in \mathbb{R}^{C \times N \times T_h}$ */
4: $\mathbf{H} = \text{Linear}\,(\mathbf{X})$     /* Input layer, $\mathbf{H} \in \mathbb{R}^{C \times N \times d}$ */
5: $\mathbf{SE} = \text{FFN}\,(\boldsymbol{\Sigma})$     /* $\mathbf{SE} \in \mathbb{R}^{N \times d}$ */
6: $\mathbf{SE} = \mathbf{SE}.\text{repeat}(C, 1, 1)$     /* $\mathbf{S} \in \mathbb{R}^{C \times N \times d}$ */
7: $hour, day, mon = \text{time\_feature}(t)$     /* Obtain hour, month and day from $t$ */
8: $\mathbf{T} = \mathbf{T}[hour].\text{repeat}(C, N, 1)$
9: $\mathbf{D} = \mathbf{D}[day].\text{repeat}(C, N, 1)$
10: $\mathbf{M} = \mathbf{M}[mon].\text{repeat}(C, N, 1)$
11: $\mathbf{Z}_0 = \mathbf{H} + \mathbf{SE} + \mathbf{T} + \mathbf{D} + \mathbf{M}$     /* $\mathbf{Z}_0 \in \mathbb{R}^{C \times N \times d}$ */
    /* MLP encoder */
12: **for** $l$ **in** $\{0, 1, \cdots, L-1\}$ **do**
13:     $\mathbf{Z}_{l+1} = \text{FFN}_l\,(\mathbf{Z}_l) + \mathbf{Z}_l$
14: **end for**
15: $\mathbf{Y} = \text{Linear}\,(\mathbf{Z}_L)$     /* Regression layer, $\mathbf{Y} \in \mathbb{R}^{C \times N \times T_f}$ */
16: $\mathbf{Y} = \mathbf{Y}.\text{transpose}\,(1, -1)$
17: **return** $\mathbf{Y}$

---

31 December 2020. Please note that these datasets are rescaled (multiplied ten times) from the raw datasets.

- **ChinaWind and ChinaTemp** are also collected from NCEI[3]. These datasets contain the daily averaged wind speed and temperature of 396 stations in China (382 stations for **Temp_CN** due to missing values), spanning 10 years from 1 January 2013 to 31 December 2022.

- **ChinaPM2.5** is collected from CNEMC[4]. It contains the hourly averaged wind speed of 1,316 stations in China, spanning 4 years from 1 January 2020 to 31 December 2024. The original dataset only provides the latitudes and longitudes of stations and we obtain the elevations of stations through Open-Elevation[5].

The statistics of the datasets are shown in Table 5 and the station distributions are shown in Figure 8.

Table 5: Statistics of datasets.

| DATASET | COVERAGE | STATION NUM | SAMPLE RATE | TIME SPAN | LENGTH |
|---|---|---|---|---|---|
| GlobalWind | Global | 3,850 | 1 hour | 2 years | 17,544 |
| GlobalTemp | Global | 3,850 | 1 hour | 2 years | 17,544 |
| ChinaWind | National | 396 | 1 day | 10 years | 3,652 |
| ChinaTemp | National | 382 | 1 day | 10 years | 3,652 |
| ChinaPM2.5 | National | 1,316 | 1 hour | 5 years | 43,539 |

## C.2 Introduction to Baselines

- **HI** [5], short for historical inertia, is a simple baseline that adopts the most recent historical data as the prediction results.

- **ARIMA** [33], short for autoregressive integrated moving average, is a statistical forecasting method that uses the combination of historical values to predict future values.

- **AGCRN** [1] is a STGNN that integrates adaptive graph convolution and recurrent networks to dynamically capture spatiotemporal dependencies in multivariate time series.

---

[3] https://www.ncei.noaa.gov/data/global-summary-of-the-day/access
[4] https://air.cnemc.cn
[5] https://open-elevation.com

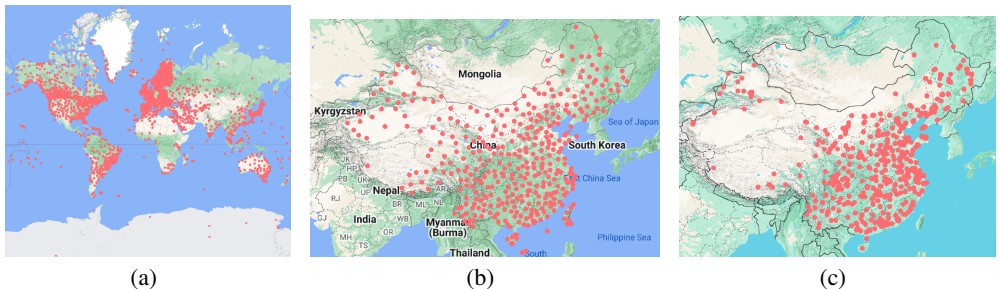

Figure 8: Distributions of the stations. (a) GlobalWind and GlobalTemp. (b) ChinaWind and ChinaTemp. (c) ChinaPM2.5.

- **MTGNN** [41] is a GNN-based model for multivariate time series forecasting. It can automatically learn the hidden spatial dependencies among variables.
- **GTS** [28] is a STGNN that learns the structure simultaneously with the GNN when the graph is unknown.
- **Informer** [53] is a Transformer for time series forecasting (TSF) with a sparse self-attention mechanism.
- **FEDformer** [54] is a frequency-enhanced Transformer combined with seasonal-trend decomposition to capture the overall trend of time series.
- **DSformer** [45] utilizes double sampling blocks to model both local and global information.
- **PatchTST** [23] divides the input time series into patches, which serve as input tokens of Transformer.
- **iTransformer** [20] is a Transformer for TSF that simply applies the attention and Feed-Forward Network (FFN) on the inverted dimensions, thereby enhancing the Transformer's capability to capture multivariate correlations.
- **DUET** [26] is a dual clustering enhanced model for TSF that clusters temporal and channel dimensions to learn the complex correlations of time series.
- **DLinear** [49] is a lightweight baseline for TSF, which consists of a linear model and a time series decomposition module.
- **N-BEATS** [24] utilizes backward and forward residual links and a very deep stack of fully-connected layers.
- **FITS** [43] is a lightweight baseline for TSF that employs a complex-valued linear layer to learn amplitude scaling and phase shifts, enabling interpolation in the complex frequency domain.
- **AirFormer** [16] employs a dartboard-like mapping and local windows to restrict attention to focusing solely on local information.
- **Corrformer** [40] utilizes a decomposition framework and replaces attention mechanisms with a more efficient multi-correlation mechanism.
- **MRIformer** [47] employs a hierarchical tree structure, stacking attention layers to capture correlations from multi-resolution data obtained by downsampling.

### C.3 Evaluation Metrics

The evaluation metrics we used in the paper are defined as follows.

**Mean Absolute Error (MAE)**

$$\text{MAE}(\hat{\mathbf{Y}}, \mathbf{Y}) = \frac{1}{N \cdot C \cdot T_f} \sum_{i=1}^{N} \sum_{j=1}^{C} \sum_{k=1}^{T_f} |\hat{\mathbf{Y}}_t^{i,j} - \mathbf{Y}_t^{i,j}|. \tag{24}$$

**Root Mean Square Error (RMSE)**

$$\text{RMSE}(\hat{\mathbf{Y}}, \mathbf{Y}) = \sqrt{\frac{1}{N \cdot C \cdot T_f} \sum_{i=1}^{N} \sum_{j=1}^{C} \sum_{k=1}^{T_f} \left( \hat{\mathbf{Y}}_t^{i,j} - \mathbf{Y}_t^{i,j} \right)^2}. \tag{25}$$

### C.4 Optimal Settings

For reproducibility purposes, we provide the optimal hyperparameters of STELLA on the five datasets, as illustrated in Figure 6. All experiments were performed on an NVIDIA GeForce RTX 4090 24GB GPU.

Table 6: The optimal settings of STELLA.

| Settings | | GlobalWind | GlobalTemp | ChinaWind | ChinaTemp | ChinaPM2.5 |
|---|---|---|---|---|---|---|
| Network Architecture | Hidden Dimension | 1024 | 2048 | 64 | 32 | 256 |
| | Layers of MLP | 2 | 2 | 2 | 3 | 2 |
| | Activation Function | ReLU | ReLU | ReLU | ReLU | ReLU |
| | Dropout | 0.2 | 0.2 | 0.2 | 0.2 | 0.2 |
| Optimization | Batch Size | 32 | 32 | 32 | 32 | 32 |
| | Epoch | 100 | 100 | 100 | 100 | 100 |
| | Optimizer | Adam | Adam | Adam | Adam | Adam |
| | Learning Rate | 5e-4 | 5e-4 | 5e-4 | 5e-4 | 5e-4 |
| | Weight Decay | 5e-4 | 5e-4 | 5e-4 | 5e-4 | 5e-4 |
| | LR Scheduler | MultistepLR | MultistepLR | MultistepLR | MultistepLR | MultistepLR |
| | - Milestone | [1,50] | [1,50] | [1,50] | [50] | [1,25,50] |
| | - $\gamma$ | 0.5 | 0.5 | 0.5 | 0.5 | 0.5 |

## D Additional Experimental Results

### D.1 Comparison with Numerical Methods

In this section, we compare our model with the numerical weather prediction (NWP) methods in short-term global weather forecasting tasks. Conventional NWP methods use PDEs to describe the atmospheric state transitions across grid points and solve them through numerical simulations. Currently, the ERA5 from the European Centre for Medium-Range Weather Forecasts (ECMWF) and the Global Forecast System (GFS) from NOAA are the most advanced global forecasting models. ERA5 provides gridded global forecasts at a 0.5° resolution while GFS at a 0.25° resolution.

To make the comparison practical, we utilize bilinear interpolation with height correction to obtain the results for scattered stations, which is aligned with the convention in weather forecasting [2, 40]. The results are shown in Table 7.

Both ERA5 and GFS fail to provide accurate predictions, which indicates that grid-based NWP methods are inadequate for fine-grained station-based predictions. In contrast, STELLA can accurately forecast the global weather for worldwide stations, significantly outperforming the numerical methods.

Table 7: Forecasting results from NWP methods and our model on global weather datasets.

| Methods | GlobalWind | | GlobalTemp | |
|---|---|---|---|---|
| | MSE | MAE | MSE | MAE |
| ERA5 (0.5°) | 2.606 | 1.847 | 5.298 | 3.270 |
| GFS (0.25°) | 3.161 | 2.340 | 3.864 | 2.287 |
| **STELLA** | **1.919** | **1.284** | **2.724** | **1.858** |

## D.2 Efficiency Analysis under Limited Computational Resources

In many application scenarios, training a DL model from scratch is necessary, even when computational resources are limited. However, complex model architectures come with significant costs, including hundreds of millions of parameters and extended training times, which hinder their applicability. In the era of large models [9, 29, 50, 42], this phenomenon is particularly evident. STELLA comes with an efficient solution for this. In this section, we conduct a further efficiency analysis under limited computational resources. To simulate scenarios with limited computational resources, we utilize an Intel Xeon Gold 6330 CPU to train and test models. The top five performing models on the GlobalWind dataset are selected for to experiment. Table 8 presents the training time and inference time of different models on CPU.

The following observations can be made: (1) Except for STELLA, the other top five models are all based on the Transformer architecture. (2) The training times for these models on a CPU are impractical. For instance, training Corrformer for 50 epochs would take approximately 10 months. In contrast, STELLA can be efficiently trained in a CPU environment, requiring only about 2 hours to train STELLA-*10k* for 50 epochs. Furthermore, due to its compact parameter size and straightforward computations, STELLA is highly suitable for deployment on edge devices for inference tasks.

Table 8: The training time and inference time of the top five models on CPU.

| Methods | Performance Ranking | Training Time / Epoch | Inference Time / Sample |
|---|---|---|---|
| PatchTST | 5 | 2.25h | 0.10s |
| Corrformer | 4 | 141h | 3.90s |
| iTransformer | 3 | 17.5h | 0.70s |
| AirFormer | 2 | 56.1h | 1.77s |
| **STELLA-*10k*** | 1 | **141s** | **2ms** |

# E Discussion

## E.1 Comparison between STELLA and Other Lightweight Methods

In the main text, we provide a comprehensive comparison between STELLA and Transformer-based models. Here, we present a further discussion about the distinctions between STELLA and other lightweight models in terms of both performance and efficiency. We conduct additional experiments on the GlobalWind dataset and compare STELLA to three lightweight TSF models, N-BEATS [24], DLinear [49], FITS [43]. Table 9 presents the performance-efficiency comparison results, from which we derive the following conclusions:

Table 9: The training time and inference time of the top five models on CPU.

| Methods | Performance Ranking | Params | Epoch Time | Max Mem. |
|---|---|---|---|---|
| N-BEATS | 10 | 121.78k | 37s | 2.00GB |
| DLinear | 7 | 2.35k | 27s | 1.10GB |
| FITS | 9 | 1.80k | 29s | 1.27GB |
| **STELLA-*10k*** | 1 | 9.98k | 30s | 792MB |

- **Performance.** In terms of performance, STELLA achieves the top ranking among the 17 baselines (15 of which are DL-based), demonstrating a substantial lead over other lightweight models. This superiority stems from the fact that other lightweight models fail in modeling spatial correlations. Taking DLinear as an example, its use of a single linear layer to predict data across all sites is inherently unsuitable for multi-station forecasting scenarios.

- **Efficiency.** We analyze efficiency using three metrics: parameter counts, training time per epoch, and maximum GPU memory usage. FITS and DLinear outperform STELLA in terms of parameter counts. However, STELLA achieves comparable training speed to other lightweight models, while its GPU memory usage is even *lower*. This is because other

lightweight models incorporate additional operations to balance performance. For instance, DLinear introduces convolutional modules, while FITS employs complex frequency-domain interpolation. Despite having linear layers as their backbone, these models are less efficient than anticipated. Overall, STELLA maintains a high level of efficiency without compromising performance.

### E.2 Comparison between STELLA and PINNs

While PINNs (physics-informed neural networks) often incorporate physical constraints or regularization, our approach is different in mechanism and purpose. Instead of using physical equations as loss terms or auxiliary supervision in ATSF tasks [12, 8] , we draw theoretical motivation from the governing PDEs of atmospheric dynamics to construct input representations (STPE) that inject geographical and temporal priors directly into the model. To the best of our knowledge, this is the first work to prove, both theoretically and empirically, that such a representation alone, even coupled with a simple MLP, can surpass many state-of-the-art ATSF models in both accuracy and efficiency.

### E.3 Can STPE be Applied to Linear Model?

As shown in §4.5, STPE can be applied to Transformer-based models to enhance their performance, which naturally raises a question: *Can STPE also be applied to more lightweight methods (e.g., a linear model)?*

To address this, we conducted extensive experiments and found that the improvement was relatively marginal, especially in global weather forecasting tasks (GlobalWind and GlobalTemp datasets), whereas more substantial gains were observed in national-scale tasks (ChinaWind and ChinaTemp datasets), as shown in Table 10. According to **Theorem** 1, the evolution of atmospheric states is nonlinear, thus, the fitting capability is the core limitation for linear models.

Table 10: Improvements obtained by the adoption of STPE.

| Datasets | | GlobalWind | | GlobalTemp | | ChinaWind | | ChinaTemp | |
|---|---|---|---|---|---|---|---|---|---|
| Metric | | RMSE | MAE | RMSE | MAE | RMSE | MAE | RMSE | MAE |
| DLinear | Original | 2.005 | 1.350 | 3.149 | **2.072** | 7.309 | 5.031 | 4.990 | 3.659 |
| | **+STPE** | **2.002** | **1.348** | **3.138** | 2.078 | **7.250** | **5.002** | **4.881** | **3.596** |

## F  Case Study

In the main text, we present multi-station collaborative prediction results to provide an intuitive understanding of STELLA's ability to capture spatial correlations and perform collaborative predictions. Here, to enable a clear comparison among different models, we provide supplementary prediction cases from individual stations. We select three representative datasets for each ATSF task, and the results are given by the following advanced models: STELLA, iTransformer [20], Corrformer [40], AirFormer [16], PatchTST [23], DLinear [49]. Figure 9-11 indicates that STELLA provides the most accurate prediction and demonstrates superior performance among the models.

## G  Limitations and Future Work

**Multivariate Correlations.** Atmospheric variables are tightly coupled (Eq.14); however, in our modeling, we decoupled these variables, overlooking the multivariate correlations and training separate models for each atmospheric variable. This was primarily driven by performance considerations and was consistent with previous work [40]. Accounting for multivariate correlations and jointly forecasting multiple atmospheric variables might introduce learning challenges, potentially leading to degraded prediction performance. Since the number of atmospheric variables to be predicted is usually limited, independently forecasting each variable is a cost-effective approach relative to performance improvement, especially when using the lightweight STELLA model. Nonetheless, incorporating other meteorological variables as covariates may enhance prediction performance, which we leave as future work.

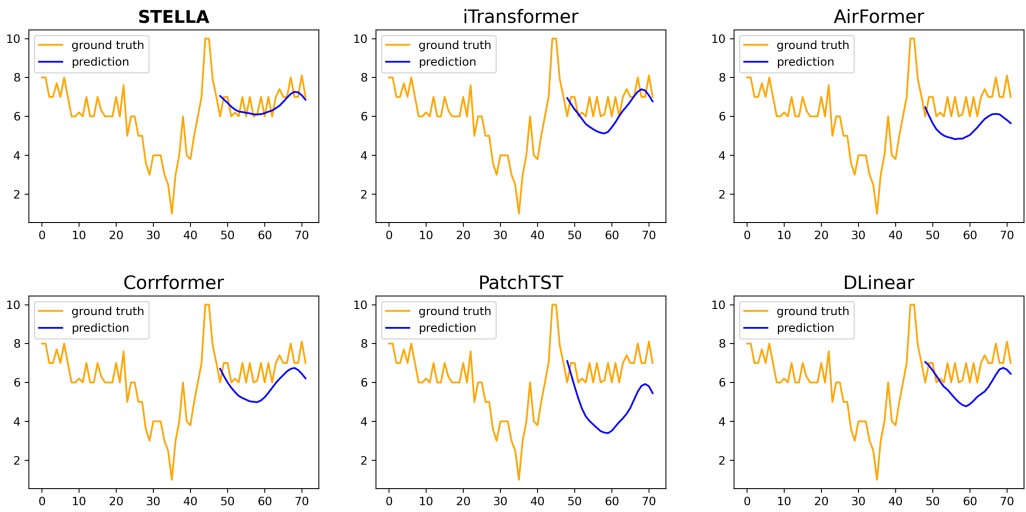

Figure 9: Visualization of the prediction results on the GlobalWind dataset.

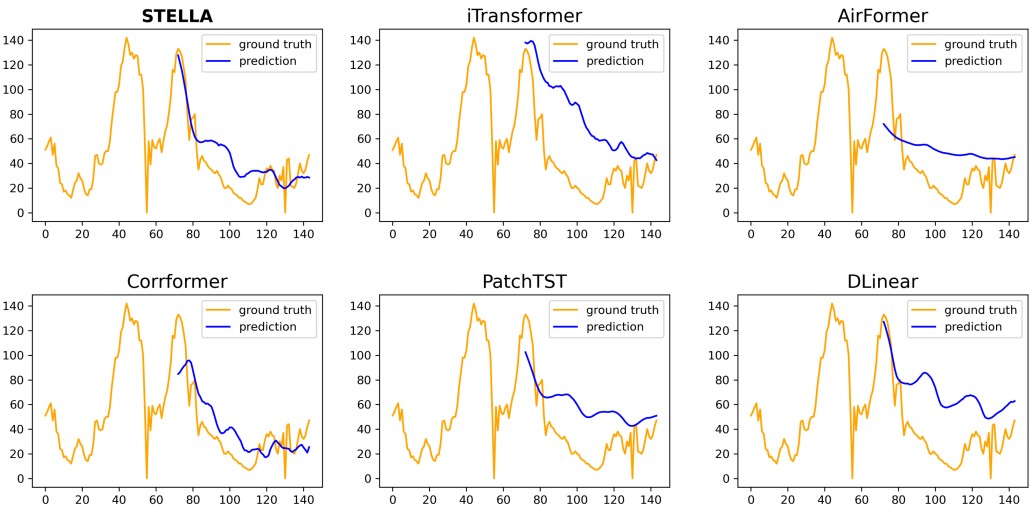

Figure 10: Visualization of the prediction results on the ChinaPM2.5 dataset.

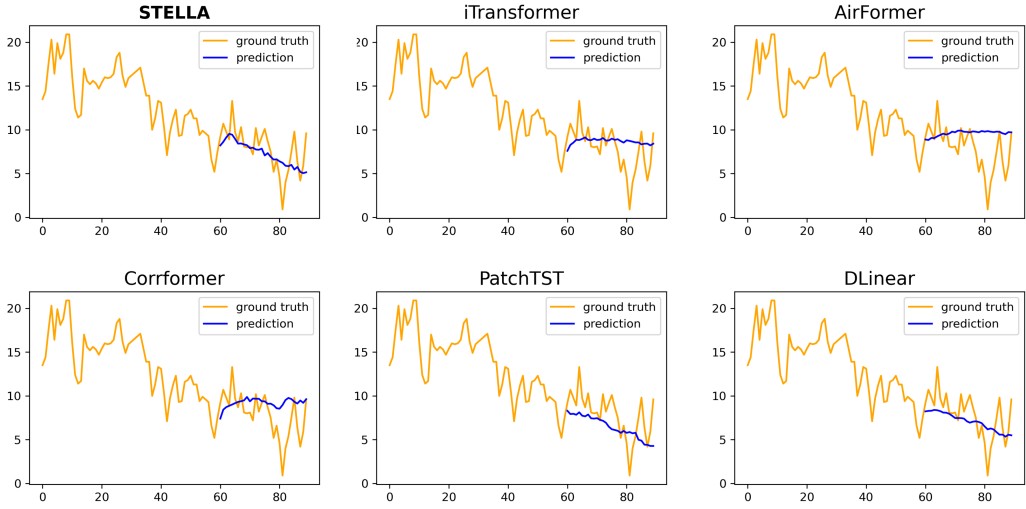

Figure 11: Visualization of the prediction results on the ChinaTemp dataset.

**Extreme Weather Forecasting.** When forecasting a target time series with violent fluctuations, the model may provide over-smooth predictions, leading to an inability to accurately forecast extreme weather events in practical applications. This is a common issue for DL models due to the use of MSE/MAE loss. A possible explanation is that MSE loss compresses the feature representations into a constrained space, limiting the model's ability to capture high-entropy features, especially those with significant variability [51]. Incorporating cross-entropy as a classification loss into the loss function may help address this issue [35]. Additionally, although we considered the physical principles of atmospheric dynamics, we did not directly incorporate physical constraints into the model's predictions. Doing so may help mitigate the issue of over-smoothing and also improve the interpretability of the predictions. We leave this as future work.

**Fitting Capacity of MLP Backbone.** We recognize that MLPs, due to their shallow and simple architecture, may face limitations in fitting capacity. We believe this constitutes the primary bottleneck limiting STELLA's further performance improvement. The motivation of this paper is to demonstrate that even minimalist architectures, when integrated with spatial-temporal knowledge, can achieve SOTA performance while offering superior efficiency. As shown in Table 4, introducing STPE into iTransformer leads to a new SOTA on GlobalWind and other datasets, demonstrating that STPE is not only effective on MLPs, but also generalizable to more expressive architectures. We leave the exploration of alternative architectures for future research.

# H  Broader Impact

This paper proposes STELLA, a lightweight approach for atmospheric time series forecasting. STELLA demonstrates high training efficiency, which helps reduce energy consumption and benefits domains such as agriculture and the economy. However, the deep learning-based approach may lack interpretability in forecasting results, and regional biases in training data may disadvantage certain populations.

