# OpenReview forum: "On the Integration of Spatial-Temporal Knowledge: A Lightweight Approach to Atmospheric Time Series Forecasting"
_NeurIPS.cc/2025/Conference — NeurIPS 2025 poster_

### Official Review · Reviewer_kupb · 2025-06-22

**Clarity:** 3
**Significance:** 2
**Originality:** 2
**Rating:** 4
**Confidence:** 5

**Summary:**

This paper presents a novel approach called STELLA (Spatial-Temporal knowledge Embedded Lightweight modeL for Atmospheric Time Series Forecasting), which challenges the prevailing assumption that complex architectures like Transformers are necessary for atmospheric time series forecasting (ATSF). The authors argue that spatial-temporal position embedding (STPE) can inherently model spatial-temporal correlations even without attention mechanisms, by integrating geographical coordinates and temporal features intrinsically linked to atmospheric dynamics. STELLA utilizes STPE and an MLP architecture instead of Transformer layers, achieving superior performance with only 10k parameters and one hour of training across five datasets compared to 16 advanced methods. The paper provides theoretical proof of STPE's effectiveness and emphasizes the importance of integrating spatial-temporal knowledge over complex model architectures.

**Questions:**

（1）The authors should provide a detailed comparison of their method with existing lightweight Transformer models and physics-integrated approaches. Highlight the unique advantages and novel contributions of STELLA in contrast to these prior works. Clarify the specific aspects in which STELLA advances the state of the art.
  (2) The authors should conduct a thorough analysis to identify the reasons for the limited performance gain.
  (3) The authors emphasize the significance of spatial - temporal correlation modeling. Existing approaches for spatial - temporal modeling typically fall into two paradigms: asynchronous spatial - temporal modeling (e.g., STTN, CLCRN) and synchronous spatial - temporal modeling (e.g., FourierGNN). The authors seem to adopt a synchronous modeling strategy. However, the paper falls short in elaborating on the details of the spatial - temporal modeling approach.

**Ethical Concerns:**

["NO or VERY MINOR ethics concerns only"]

**Final Justification:**

I thank the authors for their rebuttal, which addressed several of my concerns, and I maintain my rating of 4 (borderline accept).

**Limitations:**

yes

**Quality:**

2

**Strengths And Weaknesses:**

Strengths:
   （1）Efficiency: STELLA demonstrates significant advantages in efficiency metrics, including parameter counts, training time, and memory usage, making it suitable for environments with limited computational resources.
   （2）Theoretical Support: The authors provide theoretical analysis and proofs to support the effectiveness and efficiency of STELLA, strengthening the credibility of their claims.
Weaknesses:
   （1）The innovation of this work is insufficient. On the one hand, a great many works on lightweight Transformer schemes have already been proposed. For example, PatchTST simplifies the Transformer by introducing linear transformations, which cuts down the number of parameters and computational complexity while delivering promising performance in time series tasks. In this context, the approach of STELLA, which replaces Transformer layers with an MLP, is not sufficiently novel despite its simplicity. The authors fail to effectively highlight the unique merits of their method in contrast to other lightweight Transformer models and provide an inadequate comparison to clarify its innovative contributions. On the other hand, a significant body of research has already been carried out on the integration of physical knowledge into models. For instance, physics - guided neural networks utilize physical laws as regularizers or constraints during model training to improve the prediction of physical systems.
   （2）The model's performance improvement is rather insignificant.

---

> ### Author Rebuttal · Authors · 2025-07-30
>
> *We sincerely thank Reviewer kupb for their insightful comments. The following addresses their concerns and provides answers to their questions.*
>
> ---
>
> 1. **The innovation of our work (Weakness 1 and Question 1)**
>
>    Thank you for this valuable feedback regarding the novelty.
>
>    **(1) While we acknowledge the existence of numerous lightweight Transformers (e.g., PatchTST, iTransformer), our contribution is fundamentally distinct.** These methods typically reduce the number of tokens in attention to decrease complexity, but they continue to rely on the context aggregation over historical data. In contrast, STELLA challenges this paradigm by demonstrating that spatial-temporal correlations in ATSF can be effectively modeled *without* any attention mechanisms, purely through spatial-temporal knowledge integration via our STPE. This reframing of spatial-temporal modeling shifts the focus from complex architecture design to principled knowledge integration, which is a key novelty of our work.
>
>    **(2) While physics-guided neural networks often incorporate physical constraints or regularization, our approach is different in mechanism and purpose.** Instead of using physical equations as loss terms or auxiliary supervision [1, 2], we draw theoretical motivation from the governing PDEs of atmospheric dynamics to construct input representations (STPE) that inject geographical and temporal priors directly into the model. To the best of our knowledge, this is the first work to prove, both theoretically and empirically, that such a representation alone, even coupled with a simple MLP, can surpass many state-of-the-art ATSF models in both accuracy and efficiency.
>
>    We believe that the discussion on the distinction between lightweight Transformers and physics-guided models would further strengthen our work, and we will incorporate this into the revised version.
>
>    [1] Li, Wenyuan, et al. "Deepphysinet: Bridging deep learning and atmospheric physics for accurate and continuous weather modeling." *arXiv preprint arXiv:2401.04125* (2024).
>
>    [2] Hettige, Kethmi Hirushini, et al. "AirPhyNet: Harnessing Physics-Guided Neural Networks for Air Quality Prediction." *The Twelfth International Conference on Learning Representations*, 2024.
>
> 2. **STELLA's performance improvement (Weakness 2 and Question 2)**
>
>    We appreciate the reviewer’s concern and respectfully offer further clarification.
>
>    **(1) The performance improvements are meaningful given the complexity of ATSF tasks.** Large-scale ATSF is a particularly challenging task due to its high dimensional, non-stationary, and chaotic nature. In this context, STELLA demonstrates more significant performance gains than the performance differences between baselines. For example, on the ChinaPM2.5 dataset (Table 1), the MAE improves from 14.39 (MRIformer, 3rd best) and 14.37 (PatchTST, 2nd best) to **13.85 (STELLA_opt, 3.6%)**, highlighting that STELLA’s improvement is not marginal, but rather statistically significant given the complexity of the task.
>
>    **(2) More importantly, STELLA breaks the performance-efficiency trade-off.** A key contribution of STELLA is its ability to outperform or match SOTA models using only **~10k parameters**, orders of magnitude fewer than existing Transformers. As detailed in Table 2, STELLA is **10× to 10,000×** smaller, **100× to 300×** faster, and **~50×** more memory-efficient. Furthermore, STELLA can be easily trained on CPU environments (Table 8), while others require hours on GPUs.
>
>    **(3) We fully recognize that MLPs, due to their shallow and simple architecture, may face limitations in fitting capacity.** We believe this constitutes the primary bottleneck limiting STELLA's further performance improvement. However, our deliberate choice to pair STPE with a simple MLP was driven by a key motivation: to isolate and demonstrate the standalone power of STPE in spatial-temporal modeling. As shown in Table 4, introducing STPE into iTransformer leads to a new SOTA on GlobalWind and other datasets, demonstrating that STPE is not only effective on MLPs, but also **generalizable to more expressive architectures**. We believe our work will inspire future research to develop models with even better performance.
>
> 3. **Details of the spatial-temporal modeling approach**
>
>    Thank you for your valuable suggestion. In this paper, we categorize spatial-temporal modeling approaches into two types: (a) relative spatial-temporal modeling based on **context aggregation**, and (b) absolute spatial-temporal modeling based on **STPE**, to highlight the uniqueness of our work. As shown in Fig. 3, most existing methods rely on context aggregation (e.g., GCN modules in STGNNs or attention modules in Transformers) for spatial-temporal modeling, while this paper demonstrates that absolute spatial-temporal modeling, achieved by incorporating spatial-temporal knowledge, is a more effective and efficient approach.
>
>    Your suggestion regarding the classification of synchronous vs. asynchronous modeling is also highly insightful and will help further improve the clarity of our work. Compared to conventional synchronous approaches, asynchronous spatial-temporal modeling accounts for correlations between nodes across different timestamps. In the revised version, we will extend §2 to explicitly incorporate this taxonomy and discuss related works in both categories (e.g., STTN, CLCRN, FourierGNN).
>
> We appreciate the reviewer’s suggestion, as it will help improve the clarity and positioning of our method in the broader literature.
>
> ---
>
> *Thank you for your time and efforts. We hope this has addressed your concerns and answered your questions. Please don’t hesitate to reach out if you have any further questions.*

---

> > ### Comment · Reviewer_kupb · 2025-08-06
> >
> > Thank you to the authors for the rebuttal. I will keep my original score.

---

> > > ### Author Response · Authors · 2025-08-06
> > >
> > > Thank you for providing the insightful review, which helps us to improve the quality of our work. We will elaborate on the comparison with lightweight Transformers and physics-informed models, and expand the discussion of spatial-temporal modeling in the final version.

---

### Official Review · Reviewer_GjQt · 2025-07-02

**Clarity:** 3
**Significance:** 4
**Originality:** 3
**Rating:** 5
**Confidence:** 4

**Summary:**

The paper proposes STELLA and reveals that spatial-temporal knowlegde plays a pivotal role in atmospheric time series forecasting. The proposed method employs STPE and a simple MLP in place of Transformer layers, achieving SOTA performance with high efficiency (~10k parameters and a hour of training).

**Questions:**

Q1. In the generalization experiments, STPE are only applied to Transformer-based models. Does it maintain effectiveness when applied to other architectures such as linear models and other MLP-based models?

Q2. The theoretical analysis appears derived from PDEs of atmospheric motion. Have the authors considered implementing Neural ODEs as an alternative framework?

**Ethical Concerns:**

["NO or VERY MINOR ethics concerns only"]

**Final Justification:**

The authors' response have addressed my concerns on technical contribution description and generalization. And after reading other reviewers' comments, I would like to recommend the acceptance of this paper.

**Limitations:**

Yes

**Quality:**

3

**Strengths And Weaknesses:**

Strengths:

S1. The paper presents a novel approach to atmospheric time series forecasting by leveraging STPE with a simple MLP. The theoretical foundation is well-explained and the experimental results demonstrate STELLA's efficacy.

S2. The paper is well-organized and clearly written.

S3. The proposed model can significantly improve the efficiency and scalability to large scale ATSF tasks. The STPE module can also enhance the performance of existing models.

S4. The paper uncovers the superiority of the spatial-temporal knowledge over complex architecture, which challenges the prevailing convention that atmospheric time series forecasting necessitates intricate designs.

Weaknesses:

W1. While the paper presents an interesting finding, the technical contribution is somewhat constrained due to the adoption of well-established components (positional embedding and MLP architectures).

W2. The paper lacks an in-depth discussion on whether integrating STPE with complex architectures like Transformers and STGNNs could yield superior performance. if such combinations prove ineffective compared to MLPs, the underlying theoretical limitations should be analyzed.

---

> ### Author Rebuttal · Authors · 2025-07-30
>
> *We sincerely thank Reviewer GjQt for their insightful comments. The following addresses their concerns and provides answers to their questions.*
>
> ---
>
> 1. **Contribution of our work**
>
>    The contribution of our work lies in:
>
>    (1) **To the best of our knowledge, STELLA is the first dedicated lightweight model for ATSF. Our core innovation is uncovering the superiority of spatial-temporal knowledge over complex architecture.** While MLPs have been explored in TSF, existing methods primarily focus on historical data fitting and fail to effectively capture spatial-temporal correlations.
>
>    (2) **Theoretical contribution**: Beyond empirical results, we provide *theoretical grounding* for STPE’s effectiveness from the perspective of atmospheric dynamics (Section 3.6), which is absent in prior studies.
>
>    (3) **Practical impact**: STELLA demonstrates significant superiority in both performance and efficiency compared to other methods, making it a highly valuable solution for practical application, especially in scenarios with limited computational resources.
>
> 2. **Generalization of STPE to other models**
>
>    We notice that W2 and Q1 both concern the generalization of STPE to other models, and we address them together here.
>
>    As shown in Table 4, STPE can be effectively applied to other models to enhance their performance. Notably, when integrated with iTransformer, STPE achieves new SOTA performance on GlobalWind and other datasets. However, not all complex backbones achieve ideal results when combined with STPE, as overly complex models inherently carry overfitting risks.
>
>    STPE can also enhance the performance of other models except for Transformers. The effectiveness of STELLA demonstrates that STPE is applicable to simple *MLP architectures*. Additionally, we supplement the table below with STPE’s performance improvements on DLinear, confirming its validity for *linear models*. However, due to the inherent limitations of linear models’ fitting capacity, the performance gains are limited in complex global forecasting tasks (GlobalWind and GlobalTemp), whereas they are more significant in national tasks (ChinaWind and ChinaTemp).
>
>    |         |           | GlobalWind |           | GlobalTemp |           | ChinaWind |           | ChinaTemp |           |
>    | :------ | :-------- | :--------- | :-------- | :--------- | :-------- | :-------- | :-------- | :-------- | --------- |
>    |         |           | RMSE       | MAE       | RMSE       | MAE       | RMSE      | MAE       | RMSE      | MAE       |
>    | DLinear | w/o       | 2.005      | 1.350     | 3.149      | **2.072** | 7.309     | 5.031     | 4.990     | 3.659     |
>    |         | **+STPE** | **2.002**  | **1.348** | **3.138**  | 2.078     | **7.250** | **5.002** | **4.881** | **3.596** |
>
> 3. **Have the authors considered Neural ODEs as an alternative framework?**
>
>    Thank you for this valuable comment. Neural ODEs are indeed feasible choices, as numerous studies have already explored [1,2].  However, the key insight of our work is that integrating spatial-temporal knowledge into ATSF benefits more than employing complex architectures. This motivates us to select a sufficiently simple framework and demonstrate that it can achieve SOTA performance when integrated with spatial-temporal knowledge. In future work, we will explore more alternative frameworks, including Neural ODEs.
>
>    [1] Verma, Yogesh, Markus Heinonen, and Vikas Garg. "ClimODE: Climate and Weather Forecasting with Physics-informed Neural ODEs." *International Conference on Learning Representations*, 2024.
>
>    [2] Tian, Jindong, et al. "Air Quality Prediction with Physics-Guided Dual Neural ODEs in Open Systems." *International Conference on Learning Representations*, 2025.
>
> ---
>
> *Thank you for your time and efforts. We hope this has addressed your concerns and answered your questions. Please don’t hesitate to reach out if you have further questions or need more information.*

---

> > ### Comment · Reviewer_GjQt · 2025-08-06
> >
> > The authors' response have addressed my concerns on technical contribution description and generalization. And after reading other reviewers' comments, I would like to recommend the acceptance of this paper, so I will keep my ratings.

---

### Official Review · Reviewer_kVQ6 · 2025-07-02

**Clarity:** 3
**Significance:** 3
**Originality:** 3
**Rating:** 5
**Confidence:** 4

**Summary:**

This paper proposes STELLA, a lightweight atmospheric time series forecasting model that integrates spatial-temporal position embedding (STPE) into a simple MLP architecture. Theoretical analysis reveals that STPE can inherently capture spatial-temporal correlations, even without attention mechanisms. Extensive experiments on five real-world datasets demonstrate that STELLA achieves competitive performance with significantly fewer parameters and lower computational costs.

**Questions:**

1. Have the authors explored alternative approaches for incorporating spatial-temporal knowledge besides STPE?
2. What motivated the selection of MLP as the backbone, given its known limitations in capturing complex correlation?

**Ethical Concerns:**

["NO or VERY MINOR ethics concerns only"]

**Final Justification:**

In the rebuttal, the authors addressed all the weaknesses proposed in the review.
Now that I do not have remaining concerns,  I decide to keep the original rating after the rebuttal.

**Limitations:**

yes

**Quality:**

3

**Strengths And Weaknesses:**

Strengths:

1. This work bridges a critical gap in atmospheric time series forecasting through its incorporation of spatial coordinates and temporal information that prior works have consistently overlooked.
2. This paper proposes a simple yet effective baseline model that outperforms existing models with high efficiency.
3. The spatial-temporal position embedding (STPE) module is model-agnostic and can be integrated into other models to enhance their performance.
4. This paper provides extensive experiments and rigorous theoretical analysis to validate the effectiveness of STELLA.

Weaknesses:

1. This work focuses on spatial correlations among stations but does not explicitly account for correlations among atmospheric variables, which might further improve predictive performance.
2. While the experimental comparison covers substantial baselines, the evaluation would be strengthened by including comparisons with recent advanced time series forecasting models such as DUET [1] and FreDF [2].
3. The paper would benefit from a clearer justification for using MLP as the backbone, particularly discussing its advantages over other architectures for this specific application.

[1] Qiu X, et al. Duet: Dual clustering enhanced multivariate time series forecasting. KDD 2025.
[2] Wang H, et al. FreDF: Learning to forecast in frequency domain. ICLR 2025.

---

> ### Author Rebuttal · Authors · 2025-07-29
>
> *We sincerely thank Reviewer kvQ6 for their insightful comments. The following addresses their concerns and provides answers to their questions.*
>
> ---
>
> 1. **Correlations of atmospheric variables**
>
>    Thank you for raising this important point. Actually, Eq.(13) can be expressed as
>    $$
>    \nu_f=\mathbf A\nu_h+\mathbf F\left(\nu_h, \mu_h(\lambda,\phi,z,\tau),\lambda,\phi,z,\tau\right),
>    $$
>    where $\mu_h$ is the historical data of specific covariates correlated to $\nu$. Since $\mu$ itself is a function of $\lambda,\phi,z,\tau$, it can be treated as a hidden variable and Eq.(13) still holds. The reason for this approach is twofold: (1) Existing datasets such as GlobalWind and GlobalTemp do not include covariates. (2) Most of the previous works do not support simultaneous modeling of correlations between stations and between atmospheric variables.
>
>    The aforementioned reasons make it challenging for us to incorporate covariates into the task for evaluation. However, STELLA can incorporate the covariates. The most straightforward approach is to embed the covariates into the hidden space and add them with STPE to the embedded inputs.
>
> 2. **Advanced baselines.**
>
>    Thank you for your kind suggestions. We provide the comparison results with the advanced baselines you mentioned (Since FreDF is a training objective, we follow the original paper's configuration by using iTransformer as the backbone model). As shown in the table below, STELLA still achieves state-of-the-art performance compared to these models.
>
>    |   Model    | Metric | GlobalWind | GlobalTemp | ChinaWind | ChinaTemp | ChinaPM2.5 |
>    | :--------: | :----: | :--------: | :--------: | :-------: | :-------: | :--------: |
>    |    DUET    |  RMSE  |     -      |     -      |   7.273   |   4.480   |   26.61    |
>    |            |  MAE   |     -      |     -      |   5.041   |   3.048   |   14.41    |
>    |   FreDF    |  RMSE  |   1.971    |   2.922    |   7.224   |   4.297   |   26.93    |
>    |            |  MAE   |   1.319    |   1.861    |   5.003   |   3.191   |   14.45    |
>    | STELLA-10k |  RMSE  |   1.933    |   3.041    |   7.118   |   4.167   |   26.32    |
>    |            |  MAE   |   1.294    |   2.009    |   4.876   |   3.003   |   14.08    |
>    | STELLA-opt |  RMSE  | **1.919**  | **2.724**  | **7.104** | **4.112** | **26.15**  |
>    |            |  MAE   | **1.284**  | **1.858**  | **4.869** | **2.975** | **13.85**  |
>
>    *'-' denotes out-of-memory (OOM) errors.*
>
> 3. **Motivation of selecting MLP as backbone.**
>
>    Our key insight is that integrating spatial-temporal knowledge into ATSF benefits more than employing complex architectures. This motivates our MLP backbone choice, demonstrating that even minimalist architectures, when integrated with spatial-temporal knowledge, can achieve SOTA performance while offering superior efficiency. Alternative backbones may offer performance gains, and we hope that our work will inspire more architectures that effectively integrate spatial-temporal knowledge.
>
> 4. **Alternative approaches for spatial-temporal knowledge incorporation.**
>
>    In experiments, we explored various strategies to incorporate spatial-temporal knowledge. For example, for spatial embedding, we investigated alternative methods to encode coordinates (e.g., discretization + embedding); for temporal embedding, we experimented with encoding different time features (e.g., day-in-year). Our current choices represent the empirically optimal solutions.
>
> ---
>
> *Thank you for your time and efforts. We hope this has addressed your concerns and answered your questions. Please don’t hesitate to reach out if you have further questions or need more information.*

---

### Official Review · Reviewer_T6sb · 2025-07-05

**Clarity:** 3
**Significance:** 2
**Originality:** 3
**Rating:** 5
**Confidence:** 3

**Summary:**

This paper presents STELLA, a spatial-temporal knowledge embedded lightweight model for atmospheric time series forecasting (ATSF). STELLA utilizes spatial-temporal position embedding with a simple MLP-based architecture. The paper provides theoretical foundations for its effectiveness and computational efficiency, with empirical validation across five real-world datasets demonstrating that STELLA achieves state-of-the-art performance while maintaining high efficiency. The approach to spatial-temporal knowledge integration offers novel insights for ATSF tasks.

**Questions:**

Please refer to the proposed weakness above.

**Ethical Concerns:**

["NO or VERY MINOR ethics concerns only"]

**Final Justification:**

The author solved my main conern by providing additional experimental results. So I recommend to accpet this paper.

**Limitations:**

Please refer to above reviews.

**Quality:**

3

**Strengths And Weaknesses:**

Strengths:

1. The authors introduce a novel method, STELLA, for ATSF. The model presents state-of-the-art performance on five different weather datasets.
2. STELLA has significantly fewer parameters, shorter training times, and reduced memory usage, enhancing the scalability for large-scale forecasting.
3. The authors provide theoretical guarantees for the model's predictive performance from the perspective of atmospheric motion.

Weaknesses:

1. For efficiency analysis, what is the specific advantages of N-independent parameters compared to other models with O(N) parameters (which is deemed acceptable in my perspective)?
2. The proposed method only models the position of stations while neglects the relative spatial relationships (e.g., adjacency between stations), which show promise for boosting performance and have been explored in spatial-temporal data mining tasks.
3. The paper does not address whether alternative efficient architectures (e.g., Mamba) could attain similar results.

---

> ### Author Rebuttal · Authors · 2025-07-28
>
> *We sincerely thank Reviewer T6sb for their insightful comments. The following addresses their concerns and provides answers to their questions.*
>
> ---
>
> 1. **Advantages of N-independent parameters**
>
>    The advantages of STELLA's N-independent parameters include: (1) higher computational efficiency and faster inference speed in large-scale forecasting scenarios, and (2) constant memory footprint during deployment that remains unaffected by the number of stations, making it particularly suitable for resource-constrained environments and edge device deployment.
>
> 2. **Absolute vs. relative spatial modeling**
>
>    Thank you for raising this issue. According to Theorem 1, atmospheric variables depend on both their spatial coordinates and temporal information. Therefore, we propose that explicitly incorporating absolute spatial positions is a more effective approach for spatial modeling. In contrast, modeling adjacency relationships represents an implicit spatial modeling strategy, as the model remains unaware of actual coordinates. Furthermore, as shown in Fig. 5(b), the spatial embedding can inherently learn adjacency relationships during training.
>
> 3. **Other efficient models**
>
>    Thank you for your suggestion. In Table 1, we have compared three efficient architectures (DLinear, NBeats, and FITS). To strengthen our claim, we additionally compare 3 efficient TSF models: S-Mamba [1], SOFTS [2], and SparseTSF [3]. As shown in the table below, STELLA consistently achieves the SOTA performance, demonstrating the effectiveness of STELLA.
>
>    |            | Metric | GlobalWind | GlobalTemp | ChinaWind | ChinaTemp | ChinaPM2.5 |
>    | :--------- | :----- | :--------- | :--------- | :-------- | :-------- | :--------- |
>    | S-Mamba    | RMSE   | 1.958      | 3.001      | 7.301     | 5.012     | 26.74      |
>    |            | MAE    | 1.302      | 1.910      | 5.026     | 3.587     | 14.23      |
>    | SOFTS      | RMSE   | 1.985      | 3.057      | 7.253     | 4.962     | 26.75      |
>    |            | MAE    | 1.325      | 1.951      | 4.993     | 3.529     | 14.34      |
>    | SpaseTSF   | RMSE   | 2.117      | 3.434      | 7.230     | 5.292     | 28.71      |
>    |            | MAE    | 1.442      | 2.337      | 5.050     | 3.840     | 15.69      |
>    | STELLA-10k | RMSE   | 1.933      | 3.041      | 7.118     | 4.167     | 26.32      |
>    |            | MAE    | 1.294      | 2.009      | 4.876     | 3.003     | 14.08      |
>    | STELLA_opt | RMSE   | **1.919**  | **2.724**  | **7.104** | **4.112** | **26.15**  |
>    |            | MAE    | **1.284**  | **1.858**  | **4.869** | **2.975** | **13.85**  |
>
> ---
>
> **Reference:**
>
> [1] Wang, Zihan, et al. "Is mamba effective for time series forecasting?." *Neurocomputing* 619 (2025): 129178.
>
> [2] Han, Lu, et al. "Softs: Efficient multivariate time series forecasting with series-core fusion." *Advances in Neural Information Processing Systems* 37 (2024): 64145-64175.
>
> [3] Lin, Shengsheng, et al. "SparseTSF: modeling long-term time series forecasting with 1k parameters." *Proceedings of the 41st International Conference on Machine Learning*. 2024.
>
> ---
>
> *Thank you for your time and efforts. We hope this has addressed your concerns and answered your questions. Please don’t hesitate to reach out if you have any further questions.*

---

> > ### Comment · Reviewer_T6sb · 2025-08-09
> >
> > The authors have provided sufficient clarifications and additional evidence that adequately address my previous concerns. I am satisfied with their responses and will raise my score accordingly.

---

> > > ### Author Response · Authors · 2025-08-09
> > >
> > > We are more than happy to have addressed your concerns. Once again, thank you for your time and effort.

---

### Note · Authors · 2025-08-12

We sincerely thank all reviewers for their constructive and detailed feedback, which greatly helps us improve our work.

The reviewers generally viewed our work positively, noting it as **"a novel method"** (Reviewer T6sb, kVQ6, GjQt), **"both efficient and effective"** (Reviewer T6sb, kVQ6, GjQt, kupb), and **"bridges a critical gap in ATSF"** (Reviewer kVQ6). They also found that **theoretical grounding validates STELLA's effectiveness** (Reviewer T6sb, kVQ6, GjQt, kupb), **the STPE is model-agnostic and easy to integrate** (Reviewer kVQ6, GjQt), and **"the paper is well-written"** (Reviewer GjQt).

The final version will include the following revisions based on reviewers’ valuable feedback:

* **Efficiency analysis (Reviewer T6sb):** We have clarified that N-independent parameters yield constant deployment memory and faster inference for large-scale forecasting, suiting edge and resource-limited environments. We will further highlight our efficiency benefits in the revision.
* **Comparison with other efficient models (Reviewer T6sb) and advanced baselines (Reviewer kVQ6):** Following the reviewers' suggestion, we have added experimental results for efficient models (S-Mamba, SOFTS, SparseTSF) and advanced models (DUET， FreDF), showing that STELLA consistently outperforms them. We will include these results in the revision.
* **Motivation for selecting MLP backbone (Reviewer kVQ6, GjQt):** We have clarified in the rebuttal that our key motivation is to demonstrate STPE’s standalone power and highlight the significance of spatial-temporal knowledge. We will discuss this in the revision.
* **Generalization of STPE to other models (Reviewer GjQt):** We have added results for DLinear to demonstrate STPE’s broad applicability in our rebuttal. We will include this result and supplement experiments on more backbone models in the revision.
* **Spatial modeling taxonomy (Reviewer kupb):** We will expand Related Work to differentiate synchronous vs. asynchronous spatial-temporal modeling and position our method accordingly.
* **Comparison with lightweight Transformers and physics-guided models (Reviewer kupb):** As outlined in our rebuttal, we have clarified the key differences from these models. In the final version, we will provide a detailed comparison in Related Work to highlight STELLA’s distinctive advantages and unique contributions.

We appreciate the reviewers’ insightful suggestions, all of which will be incorporated to strengthen the final paper.

---

### Decision · Program_Chairs · 2025-09-17

**Decision:**

Accept (poster)

**Comment:**

The paper presents a spatiotemporal deep learning method for atmospheric time series forecasting. The method uses spatiotemporal positions embedding and MLPs (instead of transformer layers), improving upon established methods for atmospheric forecasting.

The reviewers praised the efficiency of the method, its theoretical guarantees and its high performance on its intended application domain. Questions arose concerning alternative efficient models (S-Mamba, SOFTS, SparseTSF), generalizability and comparisons against transformers. The authors have answered these questions, and the reviewers confirmed that their concerns were addressed for the most part. One of the reviewers (Reviewer kupb) had the lingering concern that the model could be more innovative, however the reviewer did acknowledge the considerable performance improvement and is mildly in favor of accepting the paper.

Overall, this paper represents a solid contribution to its application area (atmospheric time series forecasting) with an innovative approach and theoretical insights that might be useful in other domains as well.